# Learning Identifiable Gaussian Bayesian Networks in Polynomial Time and Sample Complexity

**Asish Ghoshal and Jean Honorio**
Department of Computer Science, Purdue University, West Lafayette, IN - 47906
{aghoshal, jhonorio}@purdue.edu

## Abstract

Learning the directed acyclic graph (DAG) structure of a Bayesian network from observational data is a notoriously difficult problem for which many non-identifiability and hardness results are known. In this paper we propose a provably polynomial-time algorithm for learning sparse Gaussian Bayesian networks with equal noise variance — a class of Bayesian networks for which the DAG structure can be uniquely identified from observational data — under high-dimensional settings. We show that $\mathcal{O}(k^4 \log p)$ number of samples suffices for our method to recover the true DAG structure with high probability, where $p$ is the number of variables and $k$ is the maximum Markov blanket size. We obtain our theoretical guarantees under a condition called *restricted strong adjacency faithfulness* (RSAF), which is strictly weaker than strong faithfulness — a condition that other methods based on conditional independence testing need for their success. The sample complexity of our method matches the information-theoretic limits in terms of the dependence on $p$. We validate our theoretical findings through synthetic experiments.

## 1 Introduction and Related Work

**Motivation.** The problem of learning the directed acyclic graph (DAG) structure of Bayesian networks (BNs) in general, and Gaussian Bayesian networks (GBNs) — or equivalently linear Gaussian structural equation models (SEMs) — in particular, from observational data has a long history in the statistics and machine learning community. This is, in part, motivated by the desire to uncover causal relationships between entities in domains as diverse as finance, genetics, medicine, neuroscience and artificial intelligence, to name a few. Although in general, the DAG structure of a GBN or linear Gaussian SEM cannot be uniquely identified from purely observational data (i.e., multiple structures can encode the same conditional independence relationships present in the observed data set), under certain restrictions on the generative model, the DAG structure can be uniquely determined. Furthermore, the problem of learning the structure of BNs exactly is known to be NP-complete even when the number of parents of a node is at most $q$, for $q > 1$, [1]. It is also known that approximating the log-likelihood to a constant factor, even when the model class is restricted to polytrees with at-most two parents per node, is NP-hard [2].

Peters and Bühlmann [3] recently showed that if the noise variances are the same, then the structure of a GBN can be uniquely identified from observational data. As observed by them, this "assumption of equal error variances seems natural for applications with variables from a similar domain and is commonly used in time series models". Unfortunately, even for the equal noise-variance case, no polynomial time algorithm is known.

**Contribution.** In this paper we develop a polynomial time algorithm for learning a subclass of BNs exactly: sparse GBNs with equal noise variance. This problem has been considered by [3] who proposed an exponential time algorithm based on $\ell_0$-penalized maximum likelihood estimation (MLE), and a heuristic greedy search method without any guarantees. Our algorithm involves estimating a $p$-dimensional inverse covariance matrix and solving $2(p - 1)$ at-most-$k$-dimensional

ordinary least squares problems, where $p$ is the number of nodes and $k$ is the maximum Markov blanket size of a variable. We show that $\mathcal{O}((k^4/\alpha^2)\log(p/\delta))$ samples suffice for our algorithm to recover the true DAG structure and to approximate the parameters to at most $\alpha$ additive error, with probability at least $1 - \delta$, for some $\delta > 0$. The sample complexity of $\mathcal{O}(k^4 \log p)$ is close to the information-theoretic limit of $\Omega(k \log p)$ for learning sparse GBNs as obtained by [4]. The main assumption under which we obtain our theoretical guarantees is a condition that we refer to as the $\alpha$-*restricted strong adjacency faithfulness* (RSAF). We show that RSAF is a strictly weaker condition than *strong faithfulness*, which methods based on independence testing require for their success. In this identifiable regime, given enough samples, our method can recover the exact DAG structure of any Gaussian distribution. However, existing exact algorithms like the PC algorithm [5] can fail to recover the correct skeleton for distributions that are not faithful, and fail to orient a number of edges that are not covered by the Meek orientation rules [6, 7]. Of independent interest is our analysis of OLS regression under the random design setting for which we obtain $\ell_\infty$ error bounds.

**Related Work.** In the this section, we first discuss some identifiability results for GBNs known in the literature and then survey relevant algorithms for learning GBNs and Gaussian SEMs.

[3] proved identifiability of distributions drawn from a restricted SEM with additive noise, where in the restricted SEM the functions are assumed to be non-linear and thrice continuously differentiable. It is also known that SEMs with linear functions and strictly non-Gaussian noise are identifiable [8]. Indentifiability of the DAG structure for the linear function and Gaussian noise case was proved by [9] when noise variables are assumed to have equal variance.

Algorithms for learning BNs typically fall into two distinct categories, namely: independence test based methods and score based methods. This dichotomy also extends to the Gaussian case. Score based methods assign a score to a candidate DAG structure based on how well it explains the observed data, and then attempt to find the highest scoring structure. Popular examples for the Gaussian distribution are the log-likelihood based BIC and AIC scores and the $\ell_0$-penalized log-likelihood score by [10]. However, given that the number of DAGs and sparse DAGs is exponential in the number of variables [4, 11], exhaustively searching for the highest scoring DAG in the combinatorial space of all DAGs, which is a feature of existing exact search based algorithms, is prohibitive for all but a few number of variables. [12] propose a score-based method, based on concave penalization of a reparameterized negative log-likelihood function, which can learn a GBN over 1000 variables in an hour. However, the resulting optimization problem is neither convex — therefore is not guaranteed to find a globally optimal solution — nor solvable in polynomial time. In light of these shortcomings, approximation algorithms have been proposed for learning BNs which can be used to learn GBNs in conjunction with a suitable score function; notable methods are Greedy Equivalence Search (GES) proposed by [13] and an LP-relaxation based method proposed by [14].

Among independence test based methods for learning GBNs, [15] extended the PC algorithm, originally proposed by [5], to learn the Markov equivalence class of GBNs from observational data. The computational complexity of the PC algorithm is bounded by $\mathcal{O}(p^k)$ with high probability, where $k$ is the maximum neighborhood size of a node, and is only efficient for learning very sparse DAGs. For the non-linear Gaussian SEM case, [3] developed a two-stage algorithm called RESIT, which works by first learning the causal ordering of the variables and then performing regressions to learn the DAG structure. As we formally show in Appendix C.1, RESIT does not work for the linear Gaussian case. Moreover, Peters et al. proved the correctness of RESIT only in the population setting. Lastly, [16] developed an algorithm, which is similar in spirit to our algorithm, for efficiently learning Poisson Bayesian networks. They exploit a property specific to the Poisson distribution called overdispersion to learn the causal ordering of variables.

Finally, the max-min hill climbing (MMHC) algorithm by [17] is a state-of-the-art hybrid algorithm for BNs that combines ideas from constraint-based and score-based learning. While MMHC works well in practice, it is inherently a heuristic algorithm and is not guaranteed to recover the true DAG structure even when it is uniquely identifiable.

## 2 Preliminaries

In this section, we formalize the problem of learning Gaussian Bayesian networks from observational data. First, we introduce some notations and definitions.

We denote the set $\{1,\ldots,p\}$ by $[p]$. Vectors and matrices are denoted by lowercase and uppercase bold faced letters respectively. Random variables (including random vectors) are denoted by italicized uppercase letters. Let $s_r, s_c \subseteq [p]$ be any two non-empty index sets. Then for any matrix $\mathbf{A} \in \mathbb{R}^{p \times p}$, we denote the $\mathbb{R}^{|s_r| \times |s_c|}$ sub-matrix, formed by selecting the $s_r$ rows and $s_c$ columns of $\mathbf{A}$ by: $\mathbf{A}_{s_r,s_c}$. With a slight abuse of notation, we will allow the index sets $s_r$ and $s_c$ to be a single index, e.g., $i$, and we will denote the index set of all row (or columns) by $*$. Thus, $\mathbf{A}_{*,i}$ and $\mathbf{A}_{i,*}$ denote the $i$-th column and row of $\mathbf{A}$ respectively. For any vector $\mathbf{v} \in \mathbb{R}^p$, we will denote its support set by: $\mathcal{S}(\mathbf{v}) = \{i \in [p] \,|\, |v_i| > 0\}$. Vector $\ell_p$-norms are denoted by $\|\cdot\|_p$. For matrices, $\|\cdot\|_p$ denotes the induced (or operator) $\ell_p$-norm and $|\cdot|_p$ denotes the element-wise $\ell_p$-norm, i.e., $|\mathbf{A}|_p \stackrel{\text{def}}{=} (\sum_{i,j} |A_{i,j}|^p)^{1/p}$. Finally, we denote the set $[p] \setminus \{i\}$ by $-i$.

Let $\mathsf{G} = (\mathsf{V}, \mathsf{E})$ be a directed acyclic graph (DAG) where the vertex set $\mathsf{V} = [p]$ and $\mathsf{E}$ is the set of directed edges, where $(i,j) \in \mathsf{E}$ implies the edge $i \leftarrow j$. We denote by $\pi_\mathsf{G}(i)$ and $\phi_\mathsf{G}(i)$ the parent set and the set of children of the $i$-th node, respectively, in the graph $\mathsf{G}$, and drop the subscript $\mathsf{G}$ when the intended graph is clear from context. A vertex $i \in [p]$ is a *terminal vertex* in $\mathsf{G}$ if $\phi_\mathsf{G}(i) = \varnothing$. For each $i \in [p]$ we have a random variable $X_i \in \mathbb{R}$, $X = (X_1, \ldots, X_p)$ is the $p$-dimensional vector of random variables, and $\mathbf{x} = (x_1, \ldots, x_p)$ is a joint assignment to $X$. Without loss of generality, we assume that $\mathbb{E}[X_i] = 0$, $\forall i \in [p]$. Every DAG $\mathsf{G} = (\mathsf{V}, \mathsf{E})$ defines a set of topological orderings $\mathcal{T}_\mathsf{G}$ over $[p]$ that are compatible with the DAG $\mathsf{G}$, i.e., $\mathcal{T}_\mathsf{G} = \{\tau \in \mathrm{S}_p \,|\, \tau(j) < \tau(i) \text{ if } (i,j) \in \mathsf{E}\}$, where $\mathrm{S}_p$ is the set of all possible permutations of $[p]$.

A Gaussian Bayesian network (GBN) is a tuple $(\mathsf{G}, \mathcal{P}(\mathsf{W}, \mathsf{S}))$, where $\mathsf{G} = (\mathsf{V}, \mathsf{E})$ is a DAG structure, $\mathsf{W} = \{w_{i,j} \in \mathbb{R} \,|\, (i,j) \in \mathsf{E} \wedge |w_{i,j}| > 0\}$ is the set of edge weights, $\mathsf{S} = \{\sigma_i^2 \in \mathbb{R}_+\}_{i=1}^p$ is the set of noise variances, and $\mathcal{P}$ is a multivariate Gaussian distribution over $X = (X_1, \ldots, X_p)$ that is *Markov* with respect to the DAG $\mathsf{G}$ and is parameterized by $\mathsf{W}$ and $\mathsf{S}$. In other words, $\mathcal{P} = \mathcal{N}(\mathbf{x}; \mathbf{0}, \boldsymbol{\Sigma})$, factorizes as follows:

$$\mathcal{P}(\mathbf{x}; \mathsf{W}, \mathsf{S}) = \prod_{i=1}^p \mathcal{P}_i(x_i; \mathbf{w}_i, \mathbf{x}_{\pi(i)}, \sigma_i^2), \tag{1}$$

$$\mathcal{P}_i(x_i; \mathbf{w}_i, \mathbf{x}_{\pi(i)}, \sigma_i^2) = \mathcal{N}(x_i; \mathbf{w}_i^T \mathbf{x}_{\pi(i)}, \sigma_i^2), \tag{2}$$

where $\mathbf{w}_i \in \mathbb{R}^{|\pi(i)|} \stackrel{\text{def}}{=} (w_{i,j})_{j \in \pi(i)}$ is the weight vector for the $i$-th node, $\mathbf{0}$ is a vector of zeros of appropriate dimension (in this case $p$), $\mathbf{x}_{\pi(i)} = \{x_j \,|\, j \in \pi(i)\}$, $\boldsymbol{\Sigma}$ is the covariance matrix for $X$, and $\mathcal{P}_i$ is the conditional distribution of $X_i$ given its parents — which is also Gaussian.

We will also extensively use an alternative, but equivalent, view of a GBN: the *linear structural equation model* (SEM). Let $\mathbf{B} = (w_{i,j} \mathbf{1}\,[(i,j) \in \mathsf{E}])_{(i,j) \in [p] \times [p]}$ be the matrix of weights created from the set of edge weights $\mathsf{W}$. A GBN $(\mathsf{G}, \mathcal{P}(\mathsf{W}, \mathsf{S}))$ corresponds to a SEM where each variable $X_i$ can be written as follows:

$$X_i = \sum_{j \in \pi(i)} B_{i,j} X_j + N_i, \;\forall i \in [p] \tag{3}$$

with $N_i \sim \mathcal{N}(0, \sigma_i^2)$ (for all $i \in [p]$) being independent noise variables and $|B_{i,j}| > 0$ for all $j \in \pi(i)$. The joint distribution of $X$ as given by the SEM corresponds to the distribution $\mathcal{P}$ in (1) and the graph associated with the SEM, where we have a directed edge $(i,j)$ if $j \in \pi(i)$, corresponds to the DAG $\mathsf{G}$. Denoting $N = (N_1, \ldots, N_p)$ as the noise vector, (3) can be rewritten in vector form as: $X = \mathbf{B}X + N$.

Given a GBN $(\mathsf{G}, \mathcal{P}(\mathsf{W}, \mathsf{S}))$, with $\mathbf{B}$ being the weight matrix corresponding to $\mathsf{W}$, we denote the *effective influence* between two nodes $i, j \in [p]$

$$\widetilde{w}_{i,j} \stackrel{\text{def}}{=} \mathbf{B}_{*,i}^T \mathbf{B}_{*,j} - B_{i,j} - B_{j,i} \tag{4}$$

The effective influence $\widetilde{w}_{i,j}$ between two nodes $i$ and $j$ is zero if: (a) $i$ and $j$ do not have an edge between them and do not have common children, or (b) $i$ and $j$ have an edge between them but the dot product between the weights to the children $(\mathbf{B}_{*,i}^T \mathbf{B}_{*,j})$ exactly equals the edge weight between $i$ and $j$ $(B_{i,j} + B_{j,i})$. The effective influence determines the Markov blanket of each node, i.e., $\forall i \in [p]$, the Markov blanket is given as: $\mathsf{S}_i = \{j \,|\, j \in -\mathsf{i} \wedge \widetilde{w}_{i,j} \neq 0\}$ [1]. Furthermore, a node is conditionally

independent of all other nodes not in its Markov blanket, i.e., $\Pr\{X_i|X_{-\mathsf{i}}\} = \Pr\{X_i|X_{\mathsf{S}_i}\}$. Next, we present a few definitions that will be useful later.

**Definition 1** (Causal Minimality [18]). *A distribution $\mathcal{P}$ is* causal minimal *with respect to a DAG structure $\mathsf{G}$ if it is not Markov with respect to a proper subgraph of $\mathsf{G}$.*

**Definition 2** (Faithfulness [5]). *Given a GBN $(\mathsf{G}, \mathcal{P})$, $\mathcal{P}$ is faithful to the DAG $\mathsf{G} = (\mathsf{V}, \mathsf{E})$ if for any $i, j \in \mathsf{V}$ and any $\mathsf{V}' \subseteq \mathsf{V} \setminus \{i, j\}$:*

$$i \text{ d-separated from } j \mid \mathsf{V}' \iff \mathrm{corr}(X_i, X_j | X_{\mathsf{V}'}) = 0,$$

*where $\mathrm{corr}(X_i, X_j|X_{\mathsf{V}'})$ is the partial correlation between $X_i$ and $X_j$ given $X_{\mathsf{V}'}$.*

**Definition 3** (Strong Faithfulness [19]). *Given a GBN $(\mathsf{G}, \mathcal{P})$ the multivariate Gaussian distribution $\mathcal{P}$ is $\lambda$-strongly faithful to the DAG $\mathsf{G}$, for some $\lambda \in (0, 1)$, if*

$$\min\{|\mathrm{corr}(X_i, X_j|X_{\mathsf{V}'})| \; : \; i \text{ is not d-separated from } j \mid \mathsf{V}', \forall i, j \in [p] \wedge \forall \mathsf{V}' \subseteq \mathsf{V} \setminus \{i, j\} \wedge \} \geq \lambda.$$

Strong faithfulness is a stronger version of the faithfulness assumption that requires that for all triples $(X_i, X_j, X_{\mathsf{V}'})$ such that $i$ is not d-separated from $j$ given $\mathsf{V}'$, the partial correlation $\mathrm{corr}(X_i, X_j|X_{\mathsf{V}'})$ is bounded away from 0. It is known that while the set of distributions $\mathcal{P}$ that are Markov to a DAG $\mathsf{G}$ but not faithful to it have Lebesgue measure zero, the set of distributions $\mathcal{P}$ that are not strongly faithful to $\mathsf{G}$ have nonzero Lebesgue measure, and in fact can be quite large [20].

The problem of learning a GBN from observational data corresponds to recovering the DAG structure $\mathsf{G}$ and parameters $\mathsf{W}$ from a matrix $\mathbf{X} \in \mathbb{R}^{n \times p}$ of $n$ i.i.d. samples drawn from $\mathcal{P}(\mathsf{W}, \mathsf{S})$. In this paper we consider the problem of learning GBNs over $p$ variables where the size of the Markov blanket of a node is at most $k$. This is in general not possible without making additional assumptions on the GBN $(\mathsf{G}, \mathcal{P}(\mathsf{W}, \mathsf{S}))$ and the distribution $\mathcal{P}$ as we describe next.

**Assumptions.** Here, we enumerate our technical assumptions.

**Assumption 1** (Causal Minimality). *Let $(\mathsf{G}, \mathcal{P}(\mathsf{W}, \mathsf{S}))$ be a GBN, then $\forall w_{i,j} \in \mathsf{W}, |w_{i,j}| > 0$.*

The above assumption ensures that all edge weights are strictly nonzero, which results in each variable $X_i$ being a non-constant function of its parents $X_{\pi(i)}$. Given Assumption 1, the distribution $\mathcal{P}$ is causal minimal with respect to $\mathsf{G}$ [3] and therefore identifiable under equal noise variances [9], i.e., $\sigma_1 = \ldots = \sigma_p = \sigma$. Throughout the rest of the paper, we will denote such Bayesian networks by $(\mathsf{G}, \mathcal{P}(\mathsf{W}, \sigma^2))$.

**Assumption 2** (Restricted Strong Adjacency Faithfulness). *Let $(\mathsf{G}, \mathcal{P}(\mathsf{W}, \sigma^2))$ be a GBN with $\mathsf{G} = (\mathsf{V}, \mathsf{E})$. For every $\tau \in \mathcal{T}_{\mathsf{G}}$, consider the sequence of graphs $\mathsf{G}[m, \tau] = (\mathsf{V}[m, \tau], \mathsf{E}[m, \tau])$ indexed by $(m, \tau)$, where $\mathsf{G}[m, \tau]$ is the induced subgraph of $\mathsf{G}$ over the first $m$ vertices in the topological ordering $\tau$, i.e., $\mathsf{V}[m, \tau] \stackrel{\mathrm{def}}{=} \{i \in [p] \mid \tau(i) \leq m\}$ and $\mathsf{E}[m, \tau] \stackrel{\mathrm{def}}{=} \{(i, j) \in \mathsf{E} \mid i \in \mathsf{V}[m, \tau] \wedge j \in \mathsf{V}[m, \tau]\}$. The multivariate Gaussian distribution $\mathcal{P}$ is restricted $\alpha$-strongly adjacency faithful to $\mathsf{G}$, provided that:*

$$(i) \; \min\{|w_{i,j}| \mid (i, j) \in \mathsf{E}\} > 3\alpha,$$

$$(ii) \; |\widetilde{w}_{i,j}| > \frac{3\alpha}{\kappa(\alpha)}, \; \forall i \in \mathsf{V}[m, \tau] \wedge j \in \mathsf{S}_i[m, \tau] \wedge m \in [p] \wedge \tau \in \mathcal{T}_{\mathsf{G}},$$

*where $\alpha > 0$ is a constant, $\widetilde{w}_{i,j}$ is the effective influence between $i$ and $j$ in the induced subgraph $\mathsf{G}[m, \tau]$ as defined in (4), and $\mathsf{S}_i[m, \tau]$ denotes the Markov blanket of node $i$ in $\mathsf{G}[m, \tau]$. The constant $\kappa(\alpha) = 1 - {2}/{(1+9|\phi_{\mathsf{G}[m,\tau]}(i)|\alpha^2)}$ if $i$ is a non-terminal vertex in $\mathsf{G}[m, \tau]$, where $|\phi_{\mathsf{G}[m,\tau]}(i)|$ is the number of children of $i$ in $\mathsf{G}[m, \tau]$, and $\kappa(\alpha) = 1$ if $i$ is a terminal vertex.*

Simply stated, the RSAF assumption requires that the absolute value of the edge weights are at least $3\alpha$ and the absolute value of the effective influence between two nodes, whenever it is non-zero, is at least $3\alpha$ for terminal nodes and ${3\alpha}/{\kappa(\alpha)}$ for non-terminal nodes. Moreover, the above should hold not only for the original DAG, but also for each DAG obtained by sequentially removing terminal vertices. The constant $\alpha$ is related to the statistical error and decays as $\Omega(k^2 \sqrt{\log p / n})$. Note that in

---

Both the definitions are equivalent under faithfulness. However, since we allow non-faithful distributions, our definition of Markov blanket is more appropriate.

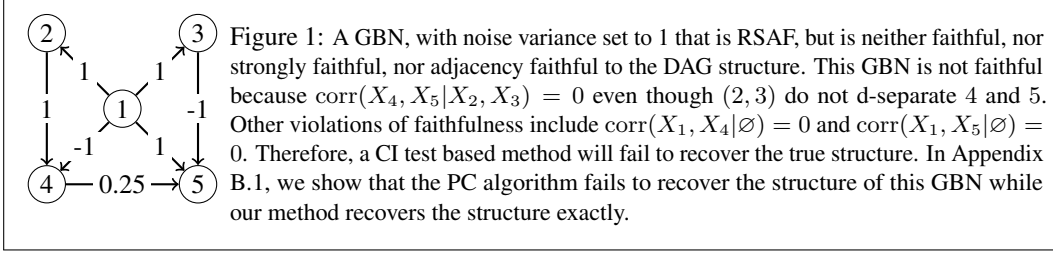

Figure 1: A GBN, with noise variance set to 1 that is RSAF, but is neither faithful, nor strongly faithful, nor adjacency faithful to the DAG structure. This GBN is not faithful because $\text{corr}(X_4, X_5 | X_2, X_3) = 0$ even though $(2,3)$ do not d-separate 4 and 5. Other violations of faithfulness include $\text{corr}(X_1, X_4 | \varnothing) = 0$ and $\text{corr}(X_1, X_5 | \varnothing) = 0$. Therefore, a CI test based method will fail to recover the true structure. In Appendix B.1, we show that the PC algorithm fails to recover the structure of this GBN while our method recovers the structure exactly.

the regime $\alpha \in (0, 1/3\sqrt{|\phi_{\mathsf{G}[m,\tau]}(i)|})$, which happens for sufficiently large $n$, then the condition on $\widetilde{w}_{i,j}$ is satisfied trivially. As we will show later, Assumption 2 is equivalent to the following, for some constant $\alpha'$,

$$\min\{|\text{corr}(X_i, X_j | X_{\mathsf{V}[m,\tau]\setminus\{i,j\}})| \mid i \in \mathsf{V}[m,\tau] \wedge j \in \mathsf{S}_i[m,\tau] \wedge m \in [p] \wedge \tau \in \mathcal{T}_{\mathsf{G}}\} \geq \alpha'.$$

At this point, it is worthwhile to compare our assumptions with those made by other methods for learning GBNs. Methods based on conditional independence (CI) tests, e.g., the PC algorithm for learning the equivalence class of GBNs developed by [15], require strong faithfulness. While strong faithfulness requires that for a node pair $(i, j)$ that are adjacent in the DAG, the partial correlation $\text{corr}(X_i, X_j | X_{\mathsf{S}})$ is bounded away from zero for all sets $\mathsf{S} \in \{\mathsf{S} \subseteq [p] \setminus \{i, j\}\}$, RSAF only requires non-zero partial correlations with respect to a subset of sets in $\{\mathsf{S} \subseteq [p] \setminus \{i, j\}\}$. Thus, RSAF is strictly weaker than strong faithfulness. The number of non-zero partial correlations needed by RSAF is also strictly a subset of those needed by the faithfulness condition. Figure 1 shows a GBN which is RSAF but neither faithful, nor strongly faithful, nor adjacency faithful (see [20] for a definition).

We conclude this section with one last remark. At first glance, it might appear that the assumption of equal variance together with our assumptions implies a simple causal ordering of variables in which the marginal variance of the variables increases strictly monotonically with the causal ordering. However, this is not the case. For instance, in the GBN shown in Figure 1 the marginal variance of the causally ordered nodes $(1, 2, 3, 4, 5)$ is $(1, 2, 2, 2, 2.125)$. We also perform extensive simulation experiments to further investigate this case in Appendix B.6.

## 3 Results

We start by characterizing the covariance and precision matrix of a GBN $(\mathsf{G}, \mathcal{P}(\mathsf{W}, \sigma^2))$. Let $\mathbf{B}$ be the weight matrix corresponding to the edge weights $\mathsf{W}$, then from (3) it follows that the covariance and precision matrix are, respectively:

$$\mathbf{\Sigma} = \sigma^2 (\mathbf{I} - \mathbf{B})^{-1} (\mathbf{I} - \mathbf{B})^{-T}, \qquad \mathbf{\Omega} = \frac{1}{\sigma^2} (\mathbf{I} - \mathbf{B})^T (\mathbf{I} - \mathbf{B}), \qquad (5)$$

where $\mathbf{I}$ is the $p \times p$ identity matrix.

**Remark 1.** *Since the elements of the inverse covariance matrix are related to the partial correlations as follows:* $\text{corr}(X_i, X_j | X_{\mathsf{V}\setminus\{i,j\}}) = -\Omega_{i,j}/\sqrt{\Omega_{i,i}\Omega_{j,j}}$. *We have that,* $|\widetilde{w}_{i,j}| \geq c\alpha$, *for some constant* $c$ *(Assumption 2), implies that* $|\text{corr}(X_i, X_j | X_{\mathsf{V}\setminus\{i,j\}})| \geq c\alpha/\sqrt{\Omega_{i,i}\Omega_{j,j}} > 0$.

Next, we describe a key property of homoscedastic noise GBNs in the lemma below, which will be the driving force behind our algorithm.

**Lemma 1.** *Let* $(\mathsf{G}, \mathcal{P}(\mathsf{W}, \sigma^2))$ *be a GBN, with* $\mathbf{\Omega}$ *being the inverse covariance matrix over* $X$ *and* $\boldsymbol{\theta}_i \overset{\text{def}}{=} \mathbb{E}[X_i | (X_{-i} = \mathbf{x}_{-i})] = \boldsymbol{\theta}_i^T \mathbf{x}_{-i}$ *being the* $i$*-th regression coefficient. Under Assumption 1, we have that*

$$i \text{ is a terminal vertex in } \mathsf{G} \iff \theta_{ij} = -\sigma^2 \Omega_{i,j}, \, \forall j \in -\mathsf{i}.$$

Detailed proofs can be found in Appendix A in the supplementary material. Lemma 1 states that, in the population setting, one can identify the terminal vertex, and therefore the causal ordering, just by assuming causal minimality (Assumption 1). However, to identify terminal vertices from a finite number of samples, one needs additional assumptions. We use Lemma 1 to develop our algorithm for learning GBNs which, at a high level, works as follows. Given data $\mathbf{X}$ drawn from a GBN, we

first estimate the inverse covariance matrix $\widehat{\boldsymbol{\Omega}}$. Then we perform a series of ordinary least squares (OLS) regressions to compute the estimators $\widehat{\boldsymbol{\theta}}_i \,\forall i \in [p]$. We then identify terminal vertices using the property described in Lemma 1 and remove the corresponding variables (columns) from $\mathbf{X}$. We repeat the process of identifying and removing terminal vertices and obtain the causal ordering of vertices. Then, we perform a final set of OLS regressions to learn the structure and parameters of the DAG.

The two main operations performed by our algorithm are: (a) estimating the inverse covariance matrix, and (b) estimating the regression coefficients $\boldsymbol{\theta}_i$. In what follows, we discuss these two steps in more detail and obtain theoretical guarantees for our algorithm.

**Inverse covariance matrix estimation.** The first part of our algorithm requires an estimate $\widehat{\boldsymbol{\Omega}}$ of the true inverse covariance matrix $\boldsymbol{\Omega}^*$. Due in part to its role in undirected graphical model selection, the problem of inverse covariance matrix estimation has received significant attention over the years. A popular approach for inverse covariance estimation, under high-dimensional settings, is the $\ell_1$-penalized Gaussian MLE studied by [21–28], among others. While, technically, these algorithms can be used in the first phase of our algorithm to estimate the inverse covariance matrix, in this paper, we use the method called CLIME, developed by Cai et. al. [29], since its theoretical guarantees do not require a quite restrictive edge-based mutual incoherence condition as in [24]. Further, CLIME is computationally attractive because it computes $\widehat{\boldsymbol{\Omega}}$ columnwise by solving $p$ independent linear programs. Even though the CLIME estimator $\widehat{\boldsymbol{\Omega}}$ is not guaranteed to be positive-definite (it is positive-definite with high probability) it is suitable for our purpose since we use $\widehat{\boldsymbol{\Omega}}$ only for identifying terminal vertices. Next, we briefly describe the CLIME method for inverse covariance estimation and instantiate the theoretical results of [29] for our purpose.

The CLIME estimator $\widehat{\boldsymbol{\Omega}}$ is obtained as follows. First, we compute a potentially non-symmetric estimate $\bar{\boldsymbol{\Omega}} = (\bar{\omega}_{i,j})$ by solving the following:

$$\bar{\boldsymbol{\Omega}} = \operatorname*{argmin}_{\boldsymbol{\Omega} \in \mathbb{R}^{p \times p}} |\boldsymbol{\Omega}|_1 \text{ s.t. } |\boldsymbol{\Sigma}^n \boldsymbol{\Omega} - \mathbf{I}|_\infty \leq \lambda_n, \tag{6}$$

where $\lambda_n > 0$ is the regularization parameter, $\boldsymbol{\Sigma}^n \stackrel{\text{def}}{=} (1/n)\mathbf{X}^T\mathbf{X}$ is the empirical covariance matrix. Finally, the symmetric estimator is obtained by selecting the smaller entry among $\bar{\omega}_{i,j}$ and $\bar{\omega}_{j,i}$, i.e., $\widehat{\boldsymbol{\Omega}} = (\widehat{\omega}_{i,j})$, where $\widehat{\omega}_{i,j} = \bar{\omega}_{i,j}\mathbf{1}\left[|\bar{\omega}_{i,j}| < |\bar{\omega}_{j,i}|\right] + \bar{\omega}_{j,i}\mathbf{1}\left[|\bar{\omega}_{j,i}| \leq |\bar{\omega}_{i,j}|\right]$. It is easy to see that (6) can be decomposed into $p$ linear programs as follows. Let $\bar{\boldsymbol{\Omega}} = (\bar{\boldsymbol{\omega}}_1, \dots, \bar{\boldsymbol{\omega}}_p)$, then

$$\bar{\boldsymbol{\omega}}_i = \operatorname*{argmin}_{\boldsymbol{\omega} \in \mathbb{R}^p} \|\boldsymbol{\omega}\|_1 \text{ s.t. } |\boldsymbol{\Sigma}^n \boldsymbol{\omega} - \mathbf{e}_i|_\infty \leq \lambda_n, \tag{7}$$

where $\mathbf{e}_i = (e_{i,j})$ such that $e_{i,j} = 1$ for $j = i$ and $e_{i,j} = 0$ otherwise. The following lemma which follows from the results of [29] and [24], bounds the maximum elementwise difference between $\widehat{\boldsymbol{\Omega}}$ and the true precision matrix $\boldsymbol{\Omega}^*$.

**Lemma 2.** *Let* $(\mathsf{G}^*, \mathcal{P}(\mathsf{W}^*, \sigma^2))$ *be a GBN satisfying Assumption 1, with* $\boldsymbol{\Sigma}^*$ *and* $\boldsymbol{\Omega}^*$ *being the "true" covariance and inverse covariance matrix over $X$, respectively. Given a data matrix* $\mathbf{X} \in \mathbb{R}^{n \times p}$ *of $n$ i.i.d. samples drawn from* $\mathcal{P}(\mathsf{W}^*, \sigma^2)$*, compute* $\widehat{\boldsymbol{\Omega}}$ *by solving* (6)*. Then, if the regularization parameter and number of samples satisfy:*

$$\lambda_n \geq \|\boldsymbol{\Omega}^*\|_1 \sqrt{(C_1/n) \log(4p^2/\delta)}, \; n \geq ((16\sigma^4\|\boldsymbol{\Omega}^*\|_1^4 C_1)/\alpha^2) \log((4p^2)/\delta),$$

*with probability at least* $1 - \delta$ *we have that* $|\boldsymbol{\Omega}^* - \widehat{\boldsymbol{\Omega}}|_\infty \leq \alpha/\sigma^2$*, where* $C_1 = 3200\left(\max_i(\boldsymbol{\Sigma}^*_{i,i})^2\right)$ *and* $\delta \in (0, 1)$*. Further, thresholding* $\widehat{\boldsymbol{\Omega}}$ *at the level* $4\|\boldsymbol{\Omega}^*\|_1\lambda_n$*, we have* $\mathcal{S}(\boldsymbol{\Omega}^*) = \mathcal{S}(\widehat{\boldsymbol{\Omega}})$*.*

**Remark 2.** *Note that in each column of the true precision matrix* $\boldsymbol{\Omega}^*$*, at most $k$ entries are non-zero, where $k$ is the maximum Markov blanket size of a node in* $\mathsf{G}$*. Therefore, the $\ell_1$ induced (or operator) norm* $\|\boldsymbol{\Omega}^*\|_1 = \mathcal{O}(k)$*, and the sufficient number of samples required for the estimator* $\widehat{\boldsymbol{\Omega}}$ *to be within $\alpha$ distance from* $\boldsymbol{\Omega}^*$*, elementwise, with probability at least* $1 - \delta$ *is* $\mathcal{O}((1/\alpha^2)k^4 \log(p/\delta))$*.*

**Estimating regression coefficients.** Given a GBN $(\mathsf{G}, \mathcal{P}(\mathsf{W}, \sigma^2))$ with the covariance and precision matrix over $X$ being $\boldsymbol{\Sigma}$ and $\boldsymbol{\Omega}$ respectively, the conditional distribution of $X_i$ given the variables in its Markov blanket is: $X_i | (X_{\mathsf{S}_i} = \mathbf{x}) \sim \mathcal{N}((\boldsymbol{\theta}_i)_{\mathsf{S}_i}^T\mathbf{x}, 1/\Omega_{i,i})$. Let $\boldsymbol{\theta}_{\mathsf{S}_i}^i \stackrel{\text{def}}{=} (\boldsymbol{\theta}_i)_{\mathsf{S}_i}$. This leads to the following generative model for $\mathbf{X}_{*,i}$:

$$\mathbf{X}_{*,i} = (\mathbf{X}_{*,\mathsf{S}_i})\boldsymbol{\theta}_{\mathsf{S}_i}^i + \boldsymbol{\varepsilon}_i', \tag{8}$$

where $\varepsilon_i' \sim \mathcal{N}(0, 1/\Omega_{i,i})$ and $\mathbf{X}_{l,\mathsf{S}_i} \sim \mathcal{N}(\mathbf{0}, \mathbf{\Sigma}_{\mathsf{S}_i,\mathsf{S}_i})$ for all $l \in [n]$. Therefore, for all $i \in [p]$, we obtain the estimator $\widehat{\boldsymbol{\theta}}^i_{\mathsf{S}_i}$ of $\boldsymbol{\theta}^i_{\mathsf{S}_i}$ by solving the following ordinary least squares (OLS) problem:

$$\widehat{\boldsymbol{\theta}}^i_{\mathsf{S}_i} = \underset{\boldsymbol{\beta} \in \mathbb{R}^{|\mathsf{S}_i|}}{\operatorname{argmin}} \frac{1}{2n} \|\mathbf{X}_{*,i} - (\mathbf{X}_{*,\mathsf{S}_i})\boldsymbol{\beta}\|_2^2 = (\mathbf{\Sigma}^n_{\mathsf{S}_i,\mathsf{S}_i})^{-1}\mathbf{\Sigma}^n_{\mathsf{S}_i,i} \tag{9}$$

The following lemma bounds the approximation error between the true regression coefficients and those obtained by solving the OLS problem. OLS regression has been previously analyzed by [30] under the random design setting. However, they obtain bounds on the predicion error, i.e., $(\boldsymbol{\theta}^i_{\mathsf{S}_i} - \widehat{\boldsymbol{\theta}}^i_{\mathsf{S}_i})^T\mathbf{\Sigma}^*(\boldsymbol{\theta}^i_{\mathsf{S}_i} - \widehat{\boldsymbol{\theta}}^i_{\mathsf{S}_i})$, while the following lemma bounds $\|\boldsymbol{\theta}^i_{\mathsf{S}_i} - \widehat{\boldsymbol{\theta}}^i_{\mathsf{S}_i}\|_\infty$.

**Lemma 3.** *Let* $(\mathsf{G}^*, \mathcal{P}(\mathsf{W}^*, \sigma^2))$ *be a GBN with* $\mathbf{\Sigma}^*$ *and* $\mathbf{\Omega}^*$ *being the true covariance and inverse covariance matrix over* $X$. *Let* $\mathbf{X} \in \mathbb{R}^{n \times p}$ *be the data matrix of* $n$ *i.i.d. samples drawn from* $\mathcal{P}(\mathsf{W}^*, \sigma^2)$. *Let* $\mathbb{E}\left[X_i | (X_{\mathsf{S}_i} = \mathbf{x})\right] = \mathbf{x}^T\boldsymbol{\theta}^i_{\mathsf{S}_i}$, *and let* $\widehat{\boldsymbol{\theta}}^i_{\mathsf{S}_i}$ *be the OLS solution obtained by solving* (9) *for some* $i \in [p]$. *Then, assuming* $\mathbf{\Sigma}^*$ *is non-singular, and if the number of samples satisfy:*

$$n \geq \frac{c|\mathsf{S}_i|^{3/2}(\|\boldsymbol{\theta}^i_{\mathsf{S}_i}\|_\infty + 1/|\mathsf{S}_i|)}{\lambda_{\min}(\mathbf{\Sigma}^*_{\mathsf{S}_i,\mathsf{S}_i})\alpha} \log\left(\frac{4|\mathsf{S}_i|}{\delta}\right),$$

*we have that,* $\|\boldsymbol{\theta}^i_{\mathsf{S}_i} - \widehat{\boldsymbol{\theta}}^i_{\mathsf{S}_i}\|_\infty \leq \alpha$ *with probability at least* $1 - \delta$, *for some* $\delta \in (0, 1)$, *with* $c$ *being an absolute constant.*

**Our algorithm.** Algorithm 1 presents our algorithm for learning GBNs. Throughout the algorithm we use as indices the true label of a node. We first estimate the inverse covariance matrix, $\widehat{\mathbf{\Omega}}$, in line 5. In line 7 we estimate the Markov blanket of each node. Then, we estimate $\widehat{\theta}_{i,j}$ for all $i$ and $j \in \widehat{\mathsf{S}}_i$, and compute the maximum per-node ratios $r_i = |-\widehat{\Omega}_{i,j}/\widehat{\theta}_{i,j}|$ in lines $8 - 11$. We then identify as terminal vertex the node for which $r_i$ is minimum and remove it from the collection of variables (lines 13 and 14). Each time a variable is removed, we perform a rank-1 update of the precision matrix (line 15) and also update the regression coefficients of the nodes in its Markov blanket (lines $16 - 20$). We repeat this process of identifying and removing terminal vertices until the causal order has been completely determined. Finally, we compute the DAG structure and parameters by regressing each variable against variables that are in its Markov blanket which also precede it in the causal order (lines $23 - 29$).

---

**Algorithm 1** Gaussian Bayesian network structure learning algorithm.

---

**Input:** Data matrix $\mathbf{X} \in \mathbb{R}^{n \times p}$.
**Output:** $(\widehat{\mathsf{G}}, \widehat{\mathsf{W}})$.
1: $\widehat{\mathbf{B}} \leftarrow \mathbf{0} \in \mathbb{R}^{p \times p}$.
2: $\mathbf{z} \leftarrow \varnothing, \mathbf{r} \leftarrow \varnothing$.  ▷ $\mathbf{z}$ stores the causal order.
3: $\mathsf{V} \leftarrow [p]$.  ▷ Remaining vertices.
4: $\mathbf{\Sigma}^n \leftarrow (1/n)\mathbf{X}^T\mathbf{X}$.
5: Compute $\widehat{\mathbf{\Omega}}$ using the CLIME estimator.
6: $\widehat{\mathbf{\Omega}}^0 = \widehat{\mathbf{\Omega}}$.
7: Compute $\widehat{\mathsf{S}}_i = \{j \in -i \mid |\widehat{\Omega}_{i,j}| > 0\}, \forall i \in [p]$.

8: **for** $i \in 1, \ldots, p$ **do**
9:      Compute $\widehat{\boldsymbol{\theta}}^i_{\widehat{\mathsf{S}}_i} = (\mathbf{\Sigma}^n_{\widehat{\mathsf{S}}_i,\widehat{\mathsf{S}}_i})^{-1}\mathbf{\Sigma}^n_{\widehat{\mathsf{S}}_i,i}$.
10:      $r_i \leftarrow \max\{|-\widehat{\Omega}_{i,j}/\widehat{\theta}_{i,j}| \mid j \in \widehat{\mathsf{S}}_i\}$.
11: **end for**
12: **for** $t \in 1 \ldots p - 1$ **do**
13:      $i \leftarrow \operatorname{argmin}(\mathbf{r})$. ▷ $i$ is a terminal vertex.
14:      Append $i$ to $\mathbf{z}$; $\mathsf{V} \leftarrow \mathsf{V} \setminus \{i\}$; $r_i \leftarrow +\infty$.
15:      $\widehat{\mathbf{\Omega}} \leftarrow \widehat{\mathbf{\Omega}}_{-\mathsf{i},-\mathsf{i}} - (1/\widehat{\Omega}_{i,i})(\widehat{\mathbf{\Omega}}_{-\mathsf{i},i})(\widehat{\mathbf{\Omega}}_{i,-\mathsf{i}})$.

16:      **for** $j \in \widehat{\mathsf{S}}_i$ **do**
17:          $\widehat{\mathsf{S}}_j \leftarrow \{l \neq j \mid |\widehat{\Omega}_{j,l}| > 0\}$.
18:          Compute $\widehat{\boldsymbol{\theta}}^j_{\widehat{\mathsf{S}}_j} == (\mathbf{\Sigma}^n_{\widehat{\mathsf{S}}_j,\widehat{\mathsf{S}}_j})^{-1}\mathbf{\Sigma}^n_{\widehat{\mathsf{S}}_j,j}$.
19:          $r_j \leftarrow \max\{|-\widehat{\Omega}_{j,l}/\widehat{\theta}_{j,l}| \mid l \in \widehat{\mathsf{S}}_j\}$.
20:      **end for**
21: **end for**
22: Append the remaining vertex in $\mathsf{V}$ to $\mathbf{z}$.
23: **for** $i \in 2, \ldots, p$ **do**
24:      $\widehat{\mathsf{S}}_{z_i} \leftarrow \{z_j | j \in [i-1]\} \cap \{j \in [p] \mid j \neq z_i \wedge |\widehat{\Omega}^0_{z_i,j}| > 0\}$.
25:      Compute $\widehat{\boldsymbol{\theta}} = (\mathbf{\Sigma}^n_{\widehat{\mathsf{S}}_{z_i},\widehat{\mathsf{S}}_{z_i}})^{-1}\mathbf{\Sigma}^n_{\widehat{\mathsf{S}}_{z_i},z_i}$.
26:      $\widehat{\pi}(z_i) \leftarrow \mathcal{S}(\widehat{\boldsymbol{\theta}})$.
27:      $\widehat{\mathbf{B}}_{z_i,\widehat{\pi}(z_i)} \leftarrow \widehat{\boldsymbol{\theta}}_{\widehat{\pi}(z_i)}$.
28: **end for**
29: $\widehat{\mathsf{E}} \leftarrow \{(i,j) \mid \widehat{B}_{i,j} \neq 0\}, \widehat{\mathsf{W}} \leftarrow \{\widehat{B}_{i,j} | (i,j) \in \widehat{\mathsf{E}}\}$, and $\widehat{\mathsf{G}} \leftarrow ([p], \widehat{\mathsf{E}})$.

---

In order to obtain our main result for learning GBNs we first derive the following technical lemma which states that if the data comes from a GBN that satisfies Assumptions $1 - 2$, then removing a terminal vertex results in a GBN that still satisfies Assumptions $1 - 2$.

**Lemma 4.** *Let* $(\mathsf{G}, \mathcal{P}(\mathsf{W}, \sigma^2))$ *be a GBN satisfying Assumptions 1 – 2, and let* $\mathbf{\Sigma}$, $\mathbf{\Omega}$ *be the (non-singular) covariance and precision matrix respectively. Let* $\mathbf{X} \in \mathbb{R}^{n \times p}$ *be a data matrix of* $n$ *i.i.d. samples drawn from* $\mathcal{P}(\mathsf{W}, \sigma^2)$, *and let* $i$ *be a terminal vertex in* $\mathsf{G}$. *Denote by* $\mathsf{G}' = (\mathsf{V}', \mathsf{E}')$ *and* $\mathsf{W}' = \{w_{i,j} \in \mathsf{W} \mid (i,j) \in \mathsf{E}'\}$ *the graph and set of edge weights, respectively, obtained by removing the node* $i$ *from* $\mathsf{G}$. *Then,* $\mathbf{X}_{j,-i} \sim \mathcal{P}(\mathsf{W}', \sigma^2) \; \forall j \in [n]$, *and the GBN* $(\mathsf{G}', \mathcal{P}(\mathsf{W}', \sigma^2))$ *satisfies Assumptions 1 – 2. Further, the inverse covariance matrix* $\mathbf{\Omega}'$ *and the covariance matrix* $\mathbf{\Sigma}'$ *for the GBN* $(\mathsf{G}', \mathcal{P}(\mathsf{W}', \sigma^2))$ *satisfy (respectively):* $\mathbf{\Omega}' = \mathbf{\Omega} - (1/\Omega_{i,i})\mathbf{\Omega}_{*,i}\mathbf{\Omega}_{i,*}$ *and* $\mathbf{\Sigma}' = \mathbf{\Sigma}_{-i,-i}$.

**Theorem 1.** *Let* $\widehat{\mathsf{G}} = ([p], \widehat{\mathsf{E}})$ *and* $\widehat{\mathsf{W}}$ *be the DAG and edge weights, respectively, returned by Algorithm 1. Under the assumption that the data matrix* $\mathbf{X}$ *was drawn from a GBN* $(\mathsf{G}^*, \mathcal{P}(\mathsf{W}^*, \sigma^2))$ *with* $\mathsf{G}^* = ([p], \mathsf{E}^*)$, $\mathbf{\Sigma}^*$ *and* $\mathbf{\Omega}^*$ *being the "true" covariance and inverse covariance matrix respectively, and satisfying Assumptions 1 – 2; if the regularization parameter is set according to Lemma 2, and if the number of samples satisfies the condition:*

$$n \geq c \left( \frac{\sigma^4 \|\mathbf{\Omega}^*\|_1^4 C_{\max}}{\alpha^2} + \frac{k^{(3/2)}(\widetilde{w}_{\max} + 1/k)}{C_{\min}\alpha} \right) \log\left( \frac{24 p^2 (p-1)}{\delta} \right),$$

*where* $c$ *is an absolute constant,* $\widetilde{w}_{\max} \stackrel{\text{def}}{=} \max\{|\widetilde{w}_{i,j}| \, | \, i \in \mathsf{V}[m,\tau] \wedge j \in \mathsf{S}_i[m,\tau] \wedge m \in [p] \wedge \tau \in \mathcal{T}_{\mathsf{G}}\}$ *with* $\widetilde{w}_{i,j}$ *being the effective influence between* $i$ *and* $j$ *(4),* $C_{\max} = \max_{i \in p}(\mathbf{\Sigma}_{i,i}^*)^2$, *and* $C_{\min} = \min_{i \in [p]} \lambda_{\min}(\mathbf{\Sigma}_{\mathsf{S}_i,\mathsf{S}_i}^*)$, *then,* $\widehat{\mathsf{E}} \supseteq \mathsf{E}^*$ *and* $\forall (i,j) \in \widehat{\mathsf{E}}$, $|\widehat{w}_{i,j} - w_{i,j}^*| \leq \alpha$ *with probability at least* $1 - \delta$ *for some* $\delta \in (0,1)$ *and* $\alpha > 0$. *Further, thresholding* $\widehat{\mathsf{W}}$ *at the level* $\alpha$ *we get* $\widehat{\mathsf{E}} = \mathsf{E}^*$.

The CLIME estimator of the precision matrix can be computed in polynomial time and the OLS steps take $\mathcal{O}(pk^3)$ time. Therefore our algorithm is polynomial time (please see Appendix C.2).

## 4 Experiments

In this section, we validate our theoretical findings through synthetic experiments. We use a class of Erdős-Rényi GBNs, with edge weights set to $\pm 1/2$ with probability $1/2$, and noise variance $\sigma^2 = 0.8$. For each value of $p \in \{50, 100, 150, 200\}$, we sampled 30 random GBNs and estimated the probability $\Pr\{\mathsf{G}^* = \widehat{\mathsf{G}}\}$ by computing the fraction of times the learned DAG structure $\widehat{\mathsf{G}}$ matched the true DAG structure $\mathsf{G}^*$ exactly. The number of samples was set to $Ck^2 \log p$, where $C$ was the control parameter, and $k$ was the maximum Markov blanket size (please see Appendix B.2 for more details). Figure 2 shows the results of the structure and parameter recovery experiments. We can see that the $\log p$ scaling as prescribed by Theorem 1 holds in practice.

Our method outperforms various state-of-the-art methods like PC, GES and MMHC on this class of Erdős-Rényi GBNs (Appendix B.3), works when the noise variables have unequal, but similar, variance (Appendix B.4), and also works for high-dimensional gene expression data (Appendix B.5).

**Concluding Remarks.** There are several ways of extending our current work. While the algorithm developed in the paper is specific to equal noise-variance case, we believe our theoretical analysis can be extended to the non-identifiable case to show that our algorithm, under some suitable conditions, can recover one of the Markov-equivalent DAGs. It would be also interesting to explore if some of the ideas developed herein can be extended to binary or discrete Bayesian networks.

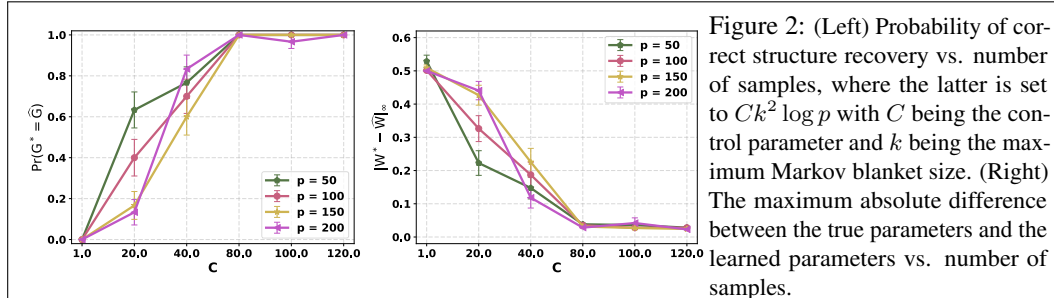

Figure 2: (Left) Probability of correct structure recovery vs. number of samples, where the latter is set to $Ck^2 \log p$ with $C$ being the control parameter and $k$ being the maximum Markov blanket size. (Right) The maximum absolute difference between the true parameters and the learned parameters vs. number of samples.

## Footnotes

[1]Our definition of Markov blanket differs from the commonly used graph-theoretic definition in that the latter includes the parents, children and *all* the co-parents of the children of node $i$ in the Markov blanket $\mathsf{S}_i$.

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
