[Supplementary Material]

# Learning Identifiable Gaussian Bayesian Networks in Polynomial Time and Sample Complexity

## Appendix A   Detailed Proofs

The following technical lemma, which characterizes the precision matrix $\boldsymbol{\Omega}$ and the conditional mean of the $i$-th random variable, given all other variables, in terms of the weight matrix $\mathbf{B}$, will be useful in later proofs.

**Lemma 5.** *Let* $(\mathsf{G}, \mathcal{P}(\mathsf{W}, \sigma^2))$ *be a GBN,* $\mathbf{B}$ *be the weight matrix corresponding to* $\mathsf{W}$ *and* $\boldsymbol{\Omega} = (\Omega_{i,j})$ *be the inverse covariance matrix over X. For all* $j \neq i$, *we have that:* $\Omega_{i,j} = (1/\sigma^2)(\mathbf{B}_{*i}^T\mathbf{B}_{*j} - B_{i,j} - B_{j,i})$, $\Omega_{i,i} = (1/\sigma^2)(1 + \mathbf{B}_{*i}^T\mathbf{B}_{*i})$ *and* $\mathbb{E}[X_i|(X_{-i} = \mathbf{x}_{-i})] = \boldsymbol{\theta}_i^T\mathbf{x}_{-i}$, *where*

$$\theta_{ij} = -\frac{\Omega_{i,j}}{\Omega_{i,i}} = \frac{B_{i,j} + B_{j,i} - \mathbf{B}_{*i}^T\mathbf{B}_{*j}}{1 + \mathbf{B}_{*i}^T\mathbf{B}_{*i}}.$$

*Proof of Lemma 5.* Consider the conditional distribution of $X_i|(X_{-i} = \mathbf{x}_{-i})$. From standard results for Gaussians (see e.g., Chapter 2 of [31]), we have that:

$$X_i|(X_{-i} = \mathbf{x}_{-i}) = \boldsymbol{\theta}_i^T\mathbf{x}_{-i} + \varepsilon_i', \text{ where} \tag{10}$$

$$\boldsymbol{\theta}_i = \boldsymbol{\Sigma}_{i,-i}(\boldsymbol{\Sigma}_{-i,-i})^{-1} = -\frac{\boldsymbol{\Omega}_{i,-i}}{\Omega_{i,i}} \text{ and } \varepsilon_i' \sim \mathcal{N}(0, \Omega_{i,i}^{-1}). \tag{11}$$

From (5) we have that:

$$\Omega_{i,j} = \frac{1}{\sigma^2}(\mathbf{I}_{*i} - \mathbf{B}_{*i})^T(\mathbf{I}_{*j} - \mathbf{B}_{*j})$$

$$= \frac{1}{\sigma^2}(\mathbf{B}_{*i}^T\mathbf{B}_{*j} - B_{i,j} - B_{j,i}) \quad (\forall j \in -i), \tag{12}$$

$$\Omega_{i,i} = \frac{1}{\sigma^2}(\mathbf{I}_{*i} - \mathbf{B}_{*i})^T(\mathbf{I}_{*i} - \mathbf{B}_{*i}) = \frac{1}{\sigma^2}(1 + \mathbf{B}_{*i}^T\mathbf{B}_{*i}), \tag{13}$$

where in (12) we used the fact that $\mathbf{I}_{*j}$ is a vector of all zeros except for a one at the $j$-th index and in (13) we used the fact that $\mathbf{B}_{*i}^T\mathbf{I}_{*i} = B_{i,i} = 0$. Combining (12) and (13) we prove our claim. $\square$

Next, we give detailed proofs of all the lemmas and theorems in our manuscript.

*Proof of Lemma 1.* The forward direction ($\Rightarrow$) follows directly from (5) and the fact that for a terminal vertex $i$, $\mathbf{B}_{*,i} = \mathbf{0}$.

Now consider the reverse direction ($\Leftarrow$). In the first case, we have $\boldsymbol{\theta}_i = -\sigma^2\boldsymbol{\Omega}_{i,*} \neq \mathbf{0}$. Then, there exists a $j \in -i$ such that $\theta_{ij} = -\sigma^2\Omega_{i,j} \neq 0$, which implies, from Lemma 5, that $\mathbf{B}_{*,i} = \mathbf{0}$ and therefore $i$ is a terminal vertex.

In the second case, we have $\boldsymbol{\theta}_i = -\sigma^2\boldsymbol{\Omega}_{i,*} = \mathbf{0}$. We will proceed with a proof by contradiction. Assume that $i$ is not a terminal vertex. Then, there exists an edge $(j, i) \in \mathsf{E}$. Further, since $\boldsymbol{\theta}_i = \mathbf{0}$, we must have, from Lemma 5, that $B_{i,j} + B_{j,i} = \mathbf{B}_{*,i}^T\mathbf{B}_{*,j} \neq 0$. Therefore, nodes $i$ and $j$ must have common children. Denote the set of common children of $i$ and $j$ by $\mathsf{C} \overset{\text{def}}{=} \phi(i) \cap \phi(j)$. There must be a node $k \in \mathsf{C}$ such that nodes $i$ and $k$ in turn do not have any common children, otherwise the DAG $\mathsf{G}$ must have a cycle. Now if $i$ and $k$ do not have any common children, then $\theta_{ik} = -\sigma^2\Omega_{i,k} \neq 0$, which is a contradiction. Therefore, $i$ must be a terminal vertex. $\square$

*Proof of Lemma 2.* From Theorem 6 of [29] we get that $|\boldsymbol{\Omega}^* - \widehat{\boldsymbol{\Omega}}|_\infty \leq 4\|\boldsymbol{\Omega}^*\|_1\lambda_n \leq \alpha/\sigma^2$, if $\lambda_n \leq \alpha/4\sigma^2\|\boldsymbol{\Omega}^*\|_1$. The lower bound requirement on $\lambda_n$ comes from Theorem 6 of [29]: $\lambda_n \geq \|\boldsymbol{\Omega}^*\|_1|\boldsymbol{\Sigma}^* - \boldsymbol{\Sigma}^n|_\infty$.

Next, we show that the empirical covariance matrix $\boldsymbol{\Sigma}^n$ is concentrated around the true covariance matrix $\boldsymbol{\Sigma}^*$, elementwise, by using the results of [24]. Note that $X_i/\sqrt{\boldsymbol{\Sigma}_{i,i}^*} \sim \mathcal{N}(0, 1)$. Therefore, from Lemma 1 of [24], we have for a fixed $i$ and $j$:

$$\Pr\{|\boldsymbol{\Sigma}_{i,j}^* - \boldsymbol{\Sigma}_{i,j}^n| \geq \varepsilon'\} \leq 4\exp\left\{\frac{-n\varepsilon'^2}{C_1}\right\}.$$

Therefore, by a union bound over all entries of $\mathbf{\Sigma}^n$, we have:

$$\implies \Pr\{|\mathbf{\Sigma}^* - \mathbf{\Sigma}^n|_\infty \le \varepsilon'\} \ge 1 - 4p^2 \exp\left\{\frac{-n\varepsilon'^2}{C_1}\right\}.$$

By setting $4p^2 \exp(-n\varepsilon'^2/C_1) = \delta$ and solving for $\varepsilon'$ we get that the following holds with probability at least $1 - \delta$:

$$|\mathbf{\Sigma}^* - \mathbf{\Sigma}^n|_\infty \le \sqrt{(C_1/n) \log\left(\frac{4p^2}{\delta}\right)}$$

The lower bound on the number of samples comes from ensuring that lower bound on $\lambda_n$ is less than the upper bound $\alpha/4\sigma^2\|\mathbf{\Omega}^*\|_1$, i.e., $\|\mathbf{\Omega}^*\|_1\sqrt{(C_1/n)\log(4p^2/\delta)} \le \alpha/4\sigma^2\|\mathbf{\Omega}^*\|_1$. $\qquad\square$

*Proof of Lemma 3.* Let $\mathbf{\Sigma}^n \overset{\mathrm{def}}{=} (1/n)\mathbf{X}^T\mathbf{X}$, be the sample covariance matrix. We first lower bound the minimum eigenvalue of the sample covariance matrix, $\lambda_{\min}(\mathbf{\Sigma}^n_{S_i,S_i})$, which will be used later on in the proof. For the purpose of this proof, we will simply write $S$ instead of $S_i$, since we will derive our results for the $i$-th node for any $i \in [p]$.

$$\lambda_{\min}(\mathbf{\Sigma}^n_{S,S}) = \min_{\|\mathbf{y}\|_2=1} \frac{1}{n}\|(\mathbf{X}_{*,S})\mathbf{y}\|_2^2 = \frac{s^2_{\min}(\mathbf{X}_{*,S})}{n}, \tag{14}$$

where $s_{\min}(.)$ (respectively $s_{\max}(.)$) denotes the minimum (respectively maximum) singular value. Now note that for any $l \in [n]$, the $|S|$-dimensional vector $\mathbf{X}_{l,S}(\mathbf{\Sigma}^*_{S,S})^{-1/2}$ is drawn from an isotropic Gaussian distribution. Therefore, from Theorem 5.39 of [32] we have:

$$s_{\min}(\mathbf{X}_{*,S}(\mathbf{\Sigma}^*_{S,S})^{-1/2}) \ge \sqrt{n} - C\sqrt{|S|} - t,$$

with probability at least $1 - 2\exp(-ct^2)$, where $C$ and $c$ are absolute constants that depend only on the sub-Gaussian norm $\|X_S(\mathbf{\Sigma}^*_{S,S})^{-1/2}\|_{\psi_2}$. Next, using the fact that $s_{\min}(\mathbf{X}_{*,S}(\mathbf{\Sigma}^*_{S,S})^{-1/2}) \le s_{\min}(\mathbf{X}_{*,S})s_{\max}((\mathbf{\Sigma}^*_{S,S})^{-1/2})$, we get:

$$s_{\min}(\mathbf{X}_{*,S}) \ge \frac{\sqrt{n} - C\sqrt{|S|} - t}{s_{\max}((\mathbf{\Sigma}^*_{S,S})^{-1/2}))}$$
$$= s_{\min}((\mathbf{\Sigma}^*_{S,S})^{1/2}))(\sqrt{n} - C\sqrt{|S|} - t). \tag{15}$$

Finally, from (14) and (15), we have that:

$$\lambda_{\min}(\mathbf{\Sigma}^n_{S,S}) \ge \lambda_{\min}(\mathbf{\Sigma}^*_{S,S})\left(1 - C\sqrt{\frac{|S|}{n}} - \frac{t}{\sqrt{n}}\right)^2$$
$$\ge \frac{\lambda_{\min}(\mathbf{\Sigma}^*_{S,S})}{4} \tag{16}$$

with probability at least $1 - 2\exp(-cn)$, where $c$ is an absolute constant, and the second line follows from controlling the second term inside the parenthesis to be at most $1/2$.

Next, from the normal equations of least squares, we have that $\widehat{\boldsymbol{\theta}}^i_S = (\mathbf{\Sigma}^n_{S,S})^{-1}\mathbf{\Sigma}^n_{S,i}$, while the true coefficient vector satisfies: $\boldsymbol{\theta}^i_S = (\mathbf{\Sigma}^*_{S,S})^{-1}\mathbf{\Sigma}^*_{S,i}$. For notational simplicity, let us write $\boldsymbol{\theta}_S$ and $\widehat{\boldsymbol{\theta}}_S$, respectively, instead of $\boldsymbol{\theta}^i_S$ and $\widehat{\boldsymbol{\theta}}^i_S$. From the entry-wise tail bounds for the sample covariance matrix derived by [24], we have that:

$$\|\mathbf{\Sigma}^*_{S,S}\boldsymbol{\theta}_S - \mathbf{\Sigma}^n_{S,S}\widehat{\boldsymbol{\theta}}_S\|_\infty = \|\mathbf{\Sigma}^*_{S,i} - \mathbf{\Sigma}^n_{S,i}\|_\infty \le \varepsilon', \tag{17}$$

with probability at least $1 - 4|S|\exp((-n\varepsilon'^2)/C_1)$. Let $\mathbf{\Delta}_S \overset{\mathrm{def}}{=} \widehat{\boldsymbol{\theta}}_S - \boldsymbol{\theta}_S$. Then, using the reverse triangle inequality we get:

$$\|\mathbf{\Sigma}^*_{S,S}\boldsymbol{\theta}_S - \mathbf{\Sigma}^n_{S,S}\widehat{\boldsymbol{\theta}}_S\|_\infty$$
$$= \|(\mathbf{\Sigma}^*_{S,S} - \mathbf{\Sigma}^n_{S,S})\boldsymbol{\theta}_S - \mathbf{\Sigma}^n_{S,S}\mathbf{\Delta}_S\|_\infty$$

$$\geq \|\mathbf{\Sigma}^n_{\mathsf{S},\mathsf{S}}\mathbf{\Delta}_\mathsf{S}\|_\infty - |\mathsf{S}|\varepsilon'\|\boldsymbol{\theta}_\mathsf{S}\|_\infty. \tag{18}$$

Next, from (17) and (18) we get:

$$\|\mathbf{\Sigma}^n_{\mathsf{S},\mathsf{S}}\mathbf{\Delta}_\mathsf{S}\|_2 \leq |\mathsf{S}|^{3/2}\varepsilon'(\|\boldsymbol{\theta}_\mathsf{S}\|_\infty + 1/|\mathsf{S}|)$$

$$\implies \|\mathbf{\Delta}_\mathsf{S}\|_2 \leq \frac{|\mathsf{S}|^{3/2}\varepsilon'(\|\boldsymbol{\theta}_\mathsf{S}\|_\infty + 1/|\mathsf{S}|)}{\lambda_{\min}(\mathbf{\Sigma}^n_{\mathsf{S},\mathsf{S}})}$$

$$\implies \|\mathbf{\Delta}_\mathsf{S}\|_\infty \leq \frac{4|\mathsf{S}|^{3/2}\varepsilon'(\|\boldsymbol{\theta}_\mathsf{S}\|_\infty + 1/|\mathsf{S}|)}{\lambda_{\min}(\mathbf{\Sigma}^*_{\mathsf{S},\mathsf{S}})} \leq \alpha,$$

with probability at least $1 - 4|\mathsf{S}|\exp\left(-\frac{n\,c\,\alpha\lambda_{\min}(\mathbf{\Sigma}^*_{\mathsf{S},\mathsf{S}})}{|\mathsf{S}|^{3/2}(\|\boldsymbol{\theta}_\mathsf{S}\|_\infty + 1/|\mathsf{S}|)}\right)$, where the second line follows from the fact that $\mathbf{\Sigma}^n_{\mathsf{S},\mathsf{S}}$ is full rank (with high probability), and the last line follows from (16) and the fact that $\|.\|_\infty \leq \|.\|_2$. Finally, by controlling the probability of error to be at most $\delta$, we derive the lower bound on the number of samples. $\qquad\square$

*Proof of Lemma 4.* Let $\mathbf{B}$ be the weight matrix corresponding to the edge weights $\mathsf{W}$, and let $\mathbf{B}' = \mathbf{B}_{-i,-i}$ denote the weight matrix corresponding to the edge weights $\mathsf{W}'$. Consider any topological order $\tau \in \mathcal{T}_\mathsf{G}$. We will denote by $(i)_\tau$ the $i$-th node in the toplogical order $\tau \in \mathcal{T}_\mathsf{G}$. The joint distribution over $(\mathbf{X}_{*,(1)_\tau}, \dots, \mathbf{X}_{*,(p)_\tau})$ is given by a linear SEM where $\mathbf{X}_{*,(i)_\tau}$ depends only on the variables occurring before the variable $(i)_\tau$ in the topological order $\tau$:

$$\mathbf{X}_{*,(i)_\tau} = \sum_{j=1}^{i-1} B_{(i)_\tau,(j)_\tau}\mathbf{X}_{*,(j)_\tau} + \varepsilon,$$

with $\varepsilon \sim \mathcal{N}(0, \sigma^2)$. Therefore, if we remove a terminal vertex, then the linear equations that describe the remaining variables do not change. Thus, if $\mathbf{\Omega}'$ and $\mathbf{\Sigma}'$ denote the precision and covariance matrix after removing node $i$, which is a terminal node, then:

$$\mathbf{\Omega}' = \frac{1}{\sigma^2}(\mathbf{I} - \mathbf{B}')^T(\mathbf{I} - \mathbf{B}')$$

$$= \frac{1}{\sigma^2}(\mathbf{I} - \mathbf{B}_{-i,-i})^T(\mathbf{I} - \mathbf{B}_{-i,-i})$$

$$\mathbf{\Sigma}' = \sigma^2(\mathbf{I} - \mathbf{B}')^{-1}(\mathbf{I} - \mathbf{B}')^{-T}$$

$$= \sigma^2(\mathbf{I} - \mathbf{B}_{-i,-i})^{-1}(\mathbf{I} - \mathbf{B}_{-i,-i})^{-T}.$$

The fact that $\mathcal{P}(\mathsf{W}', \sigma^2)$ is causal minimal (Assumption 1) and $\alpha$-RSAF (Assumption 2) is self evident. Next, using the fact that $\mathbf{\Sigma}' = \mathbf{\Sigma}_{-i,-i}$, we have:

$$0 < \lambda_{\min}(\mathbf{\Sigma}) = \min_{\{\mathbf{y}\in\mathbb{R}^p|\mathbf{y}^T\mathbf{y}=1\}} \mathbf{y}^T\mathbf{\Sigma}\mathbf{y}$$

$$\leq \min_{\{\mathbf{y}\in\mathbb{R}^p|\mathbf{y}^T\mathbf{y}=1 \wedge y_i=0\}} \mathbf{y}^T\mathbf{\Sigma}\mathbf{y}$$

$$= \min_{\mathbf{y}\in\mathbb{R}^{p-1}} \mathbf{y}^T\mathbf{\Sigma}'\mathbf{y} = \lambda_{\min}(\mathbf{\Sigma}').$$

This proves that the distribution $\mathcal{P}(\mathsf{W}', \sigma^2)$ is non-singular. Finally, the precision matrix and the covariance matrix for $X_{-i}$ is given by $\mathbf{\Omega}' = \mathbf{\Omega} - (1/\Omega_{i,i})\mathbf{\Omega}_{*,i}\mathbf{\Omega}_{i,*}$ and $\mathbf{\Sigma}' = \mathbf{\Sigma}_{-i,-i}$ respectively, which follows from standard results for marginalization of multivariate Gaussian distribution (see for instance Chapter 2 of [31]). $\qquad\square$

*Proof of Theorem 1.* First note that the lower bound on the number of samples given by Theorem 1 subsumes the sample complexity requirement of inverse covariance estimation in Lemma 2 ordinary least squares in Lemma 3. Next, by Assumption 2, we have that $\forall i \in [p]$, $\widehat{\mathsf{S}}_i = \mathsf{S}_i$, with probability at least $1 - \delta$. Therefore, from Lemma 3 $\|\boldsymbol{\theta}^i_{\mathsf{S}_i} - \widehat{\boldsymbol{\theta}}^i_{\widehat{\mathsf{S}}_i}\|_\infty \leq \alpha$, with probability at least $1 - 2\delta$.

Next, from Lemmas 2 and 3, and by our assumption that $|\widetilde{w}_{i,j}| \geq 3\alpha$, we have that for a terminal vertex $i$, the ratio $r_i$ is upper bounded as follows:

$$r_i \leq \frac{|\widetilde{w}_{i,j}| + \alpha}{\sigma^2(|\widetilde{w}_{i,j}| - \alpha)}$$

$$\leq \frac{4\alpha}{\sigma^2(2\alpha)} = \frac{2}{\sigma^2},$$

where the second line follows from the fact that $\frac{|\widetilde{w}_{i,j}|+\alpha}{\sigma^2(|\widetilde{w}_{i,j}|-\alpha)}$ is a decreasing function of $|\widetilde{w}_{i,j}|$. Similarly, if $i$ is a non-terminal vertex and has $c_i$ children, then the ratio is lower bounded as follows:

$$r_i \geq \left(\frac{1}{\sigma^2}\right) \frac{|\widetilde{w}_{i,j}| - \alpha}{\frac{|\widetilde{w}_{i,j}|}{1+\|\mathbf{w}_{*,i}\|_2^2} + \alpha}$$

In order for our algorithm to correctly identify a terminal vertex in line 13, we need to ensure that the lower bound on $r_i$ for a non-terminal vertex is strictly large than the upperbound on $r_i$ for a terminal vertex. Therefore, we need to ensure that:

$$\left(\frac{1}{\sigma^2}\right) \frac{|\widetilde{w}_{i,j}| - \alpha}{\frac{|\widetilde{w}_{i,j}|}{1+\|\mathbf{w}_{*,i}\|_2^2} + \alpha} > \frac{2}{\sigma^2}$$

Let $c_i$ be the number of children of the $i$-th node. Then, using the fact that $\|\mathbf{w}_{*,i}\|_2^2 \geq 9c_i\alpha^2$, and the function on the left hand side of the inequality above is an increasing function of $|\widetilde{w}_{i,j}|$, this further simplifies to

$$|\widetilde{w}_{i,j}| - \alpha > 2\left(\frac{|\widetilde{w}_{i,j}|}{1 + 9c_i\alpha^2} + \alpha\right)$$

$$\implies |\widetilde{w}_{i,j}| > \frac{3\alpha}{1 - \frac{2}{1+9c_i\alpha^2}}.$$

Therefore, by Assumption 2 (ii), in line 13 of Algorithm 1 we correctly identify a terminal vertex with probability at least $1 - 3\delta$. Using an union bound over the $p-1$ iterations we conclude that, with probability at least $1 - 3(p-1)\delta$, Algorithm 1 recovers a correct causal ordering of the nodes.

Next in line 25, the true coefficient vector satisfies: $\boldsymbol{\theta}^* = \boldsymbol{\Sigma}_{z_i,\widehat{\mathsf{S}}_{z_i}}(\boldsymbol{\Sigma}_{\widehat{\mathsf{S}}_{z_i},\widehat{\mathsf{S}}_{z_i}})^{-1} = \frac{\bar{\boldsymbol{\Omega}}_{z_i,\widehat{\mathsf{S}}_{z_i}}}{\bar{\Omega}_{z_i,z_i}}$, where $\bar{\boldsymbol{\Omega}}$ denotes the inverse covariance matrix over $X_{\{z_i\}\cup\widehat{\mathsf{S}}_{z_i}}$. From the fact that, a node is independent of its non-descendants given its parents, the non-zero entries of $\boldsymbol{\theta}^*$ correctly identifies the parent set of $z_i$. Therefore, by RSAF (Assumption 2), which states that the absolute value of the minimum non-zero entry in $\bar{\boldsymbol{\Omega}}$ is at least $3\alpha$, we have that the support of the OLS estimate $\widehat{\boldsymbol{\theta}}$ in line 25 correctly recovers the parent set for $z_i$ with high probability, i.e., $\Pr\{\widehat{\pi}(z_i) \neq \pi_{\mathsf{G}^*}(z_i)\} \leq 3(p-1)\delta$.

Finally, from Lemma 3 and another union bound over $p-1$ iterations of learning the parameters of the GBN, we get that $|\mathbf{B}^* - \widehat{\mathbf{B}}|_\infty \leq \alpha$ with probability at least $1 - 6(p-1)\delta$. Together with condition (i) of Assumption 2, this implies $\widehat{\mathsf{E}} = \mathsf{E}^*$ with probability at least $1 - 6(p-1)\delta$. Setting $6(p-1)\delta = \delta'$ for some $\delta' \in (0,1)$ we prove our claim. $\square$

## Appendix B Experiments

### B.1 Our method vs PC algorithm on a non-faithful GBN

We ran our method and the PC algorithm on the example given in Figure 1. We sampled 50000 samples from the GBN to ensure that the CI tests used by the PC algorithm are accurate. The following figure shows, from left to right, the true graph, the graph learned by our algorithm (with edge weights rounded to two decimal places), and the graph recovered by the PC algorithm.

## B.2 Experimental setup for structure recovery experiments

In this section, we describe in more detail our experimental setup for the various structure recovery experiments we performed. We sample a random DAG structure $\mathsf{G}^*$ over $p$ nodes by first generating an Erdős-Rényi undirected graph where each edge is sampled independently with probability $q$. Then, we randomly select a permutation of the vertex set $[p]$ and direct the edges as $i \rightarrow j$ if the node $i$ appears before node $j$ in the permutation. We then generate a GBN $(\mathsf{G}^*, \mathcal{P}(\mathsf{W}^*, \sigma^2))$ by setting the noise variance $\sigma^2 = 0.8$ for all nodes and randomly setting the edge weights to $w_{i,j}^* = \pm 1/2$ with probability $1/2$. It is easy to verify that the minimum non-zero effective influence is $0.25$ in this class of graphs. For each value of $p \in \{50, 100, 150, 200\}$, and corresponding $q \in \{0.01, 0.005, 0.0033, 0.0025\}$, we sampled 30 random GBNs and estimated the probability $\Pr\{\mathsf{G}^* = \widehat{\mathsf{G}}\}$ by computing the fraction of times the learned DAG structure $\widehat{\mathsf{G}}$ matched the true DAG structure $\mathsf{G}^*$ exactly. The number of samples was set to $Ck^2 \log p$, where $C$ was the control parameter for each experiment For the structure recovery experiments, where computed the probability of successful structure recovery as the number of samples were varied, we discarded randomly sampled GBNs for which the minimum eigenvalue of the inverse covariance matrix was less than $0.05$ to avoid numerical issues. The number of samples was set to $Ck^2 \log p$, where $C$ was the control parameter and was chosen to be in $\{1, 20, 40, 80, 100, 120\}$, and $k$ was the maximum size of the Markov blanket across all nodes in the sampled DAG $\mathsf{G}^*$. The mean and maximum value of $k$ (across 30 runs) for the different choices of $p$ was $\{3.2, 3.68, 4.12, 4.39\}$ and $\{7, 10, 7, 9\}$ respectively. The regularization parameter was set to $\lambda_n = 0.5k\sqrt{(\log p)/n}$, as prescribed by Lemma 2.

## B.3 Comparison with state-of-the-art methods

In this section, we compare the performance of our method against other state-of-the-art methods. Once again, we sampled DAGs according to the procedure described in Appendix B.2. We considered three methods for comparison: PC algorithm for learning GBNs by [15], the greedy equivalence search (GES) algorithm by [13], and the max-min hill climbing (MMHC) algorithm by [17]. The first two of the three algorithms estimate the Markov equivalence class and therefore return a completed partially directed acyclic graph (CPDAG). However, in our experiments, the sampled DAGs belong to Markov equivalence classes of size 1. Therefore, the CPDAGs should ideally have no undirected edges. The number of samples was set to $120k^2 \log p$ and the regularization parameter for our method was set to $2\sqrt{(\log p)/n}$. We do not compare against the $\ell_0$ penalized MLE algorithm by [9] for the equal variance case, which is an exact algorithm, since it searches through the *super-exponential* space of all DAGs and therefore does not scale beyond 20 nodes. The GES algorithm uses the $\ell_0$-penalized Gaussian MLE score proposed by [9] to greedily search for the best structure. We used the R package *pcalg* for the implementation of the PC and GES algorithms, and the *bnlearn* package for the implementation of the MMHC algorithm. MMHC and PC take an additional tuning parameter $\alpha$ which is the desired significance level for the individual conditional independence tests. We tested values of $\alpha \in \{0.01, 0.001, 0.0001\}$ and found that $\alpha = 0.0001$ gave the best results on an average. The number of samples was set to $120k^2 \log p$ and the regularization parameter for our method was set to $2\sqrt{(\log p)/n}$. We also used both the BIC score and the Bayesian Gaussian equivalent (BGe) score for the MMHC algorithm and found that BGe produced better results on an average. We computed the mean precision, recall, and running time in seconds, for each method, across 30 randomly sampled GBNs. Precision is defined as the fraction of all predicted (directed) edges that are actually present in the true DAG, while recall is defined as the fraction of directed edges in the true DAG that the method was able to recover. All methods were run on a single core of Intel® Xeon® running at 3.00 Ghz. Table 1 shows that our method outperforms existing methods in terms of precision and recall. Moreover, our method, is the fastest among all methods for $p \leq 100$, and is always faster than MMHC. Among, MMHC, GES and PC, the PC algorithm performed the best since it is an exact algorithm. However, the PC algorithm failed to direct many edges as is evident from its low precision score.

## B.4 Unequal noise variance

We set out to understand the performance of our algorithm when we relax the assumption of equal noise variance. Clearly, in this case, we no longer have identifiability of the true DAG structure. Therefore, we instead ask the following experimental question: "What fraction of the true edges

| Method | Precision | Recall | Seconds | Precision | Recall | Seconds |
|---|---|---|---|---|---|---|
| | p = 50 | | | p = 100 | | |
| PC | $0.587 \pm 0.015$ | $0.996 \pm 0.004$ | $0.177 \pm 0.013$ | $0.587 \pm 0.008$ | $0.999 \pm 0.001$ | $0.570 \pm 0.044$ |
| GES | $0.206 \pm 0.014$ | $0.396 \pm 0.031$ | $0.206 \pm 0.025$ | $0.204 \pm 0.013$ | $0.372 \pm 0.020$ | $0.557 \pm 0.045$ |
| MMHC | $0.581 \pm 0.038$ | $0.583 \pm 0.038$ | $0.460 \pm 0.049$ | $0.529 \pm 0.019$ | $0.533 \pm 0.019$ | $1.417 \pm 0.141$ |
| Ours | $\mathbf{1.000 \pm 0.000}$ | $\mathbf{1.000 \pm 0.000}$ | $\mathbf{0.089 \pm 0.005}$ | $\mathbf{1.000 \pm 0.000}$ | $\mathbf{1.000 \pm 0.000}$ | $\mathbf{0.534 \pm 0.004}$ |
| | p = 150 | | | p = 200 | | |
| PC | $0.572 \pm 0.006$ | $0.996 \pm 0.002$ | $1.392 \pm 0.043$ | $0.573 \pm 0.005$ | $0.997 \pm 0.001$ | $1.876 \pm 0.080$ |
| GES | $0.162 \pm 0.009$ | $0.333 \pm 0.017$ | $\mathbf{1.031 \pm 0.036}$ | $0.143 \pm 0.005$ | $0.310 \pm 0.011$ | $\mathbf{1.610 \pm 0.077}$ |
| MMHC | $0.566 \pm 0.014$ | $0.577 \pm 0.015$ | $2.934 \pm 0.241$ | $0.582 \pm 0.012$ | $0.593 \pm 0.012$ | $5.511 \pm 0.355$ |
| Ours | $\mathbf{1.000 \pm 0.000}$ | $\mathbf{1.000 \pm 0.000}$ | $1.988 \pm 0.010$ | $\mathbf{1.000 \pm 0.000}$ | $\mathbf{1.000 \pm 0.000}$ | $5.130 \pm 0.030$ |

Table 1: Performance of different algorithms across 30 randomly sampled GBNs for each value of $p \in \{50, 100, 150, 200\}$. Numbers in bold are the best for each metric across different algorithms. Our method always recovers the true DAG structure exactly.

can we recover if we perturb the noise variance of the nodes slightly?" For this experiment, we sampled GBNs as described in the previous paragraph. However, instead of setting the noise variance to be 0.8 for all nodes, we set the noise variance for each node to be one of $\{1, 1 - \gamma, 1 + \gamma\}$ with probability $1/3$, where $\gamma$ is the noise parameter. From Figure 3 we note that in the regime where the noise variance of the different nodes varies by 0.125, i.e., between 0.9375 and 1.0625, we still achieve close-to-perfect recovery.

Figure 3: Precision and Recall vs. noise parameter $\gamma$, where the noise variance for each variable was set to one of $\{1, 1 - \gamma, 1 + \gamma\}$ with equal probability. As $\gamma$ decreases, the accuracy and recall increases and we achieve perfect recovery when $\gamma = 0$, i.e. when the variables have equal noise variance.

## B.5 Experiments on real-world data

First, we test our method on extremely high-dimensional data sets. We used 14 cancer data sets publicly available at the Gene Expression Omnibus (http://www.ncbi.nlm.nih.gov/geo/). We preprocessed the data so that each variable is zero mean and unit variance across the dataset. We use the method of [33] so that the biggest connected component has at most $N' = 500$ variables (as prescribed in [33]). The technique of [33] computes a graph from edges with an absolute value of the covariance higher than the regularization parameter of the $\ell_1$-regularized MLE, and then splits the graph into its connected components. Table 2 shows the number of edges, maximum node degree, and the running time (wall clock time) of our method on 14 gene expression data sets. All experiments were run on a single core of Intel® Xeon® running at 3.00 Ghz. Existing state-of-the-art methods like PC, GES and MMHC, do not scale to such high dimensions.

Next, we qualitatively evaluate a GBN learned by our algorithm. We used gene expression data for 590 subjects with breast invasive carcinoma from the *cancer genome atlas* dataset. The dataset is publicly available at http://tcga-data.nci.nih.gov/tcga/. We used 187 genes commonly regulated in cancer that were identified on independent datasets by [34]. The genes are the following: ABCA8, ABHD6, ACLY, ADAM10, ADAM12, ADHFE1, AGXT2, ALDH6A1, ANK2, ANKS1B, ANP32E, AP1S1, APOL2, ARL4D, ARPC1B, AURKA, AYTL2, BAT2D1, BAX, BFAR, BID, BOLA2, BRP44L, C10orf116, C17orf27, C1orf58, C1orf96, C5orf4, C6orf60, C8orf76, CALU, CARD4,

| Dataset | Disease | Samples | Variables | Edges | Max degree | Time (Sec) |
|---------|---------|---------|-----------|-------|------------|------------|
| GSE1898 | Liver cancer | 182 | 21,794 | 1257 | 7 | 245.9 |
| GSE29638 | Colon cancer | 50 | 22,011 | 2182 | 8 | 112.1 |
| GSE30378 | Colon cancer | 95 | 22,011 | 1988 | 6 | 231.0 |
| GSE20194 | Breast cancer | 278 | 22,283 | 1699 | 10 | 213.2 |
| GSE22219 | Breast cancer | 216 | 24,332 | 1412 | 16 | 267.8 |
| GSE13294 | Colon cancer | 155 | 54,675 | 4325 | 10 | 346.5 |
| GSE17951 | Prostate cancer | 154 | 54,675 | 4168 | 11 | 84.0 |
| GSE18105 | Colon cancer | 111 | 54,675 | 3111 | 13 | 215.1 |
| GSE1476 | Colon cancer | 150 | 59,381 | 2056 | 9 | 211.1 |
| GSE14322 | Liver cancer | 76 | 104,702 | 3815 | 31 | 227.1 |
| GSE18638 | Colon cancer | 98 | 235,826 | 1079 | 3 | 400.7 |
| GSE33011 | Ovarian cancer | 80 | 367,657 | 5926 | 11 | 339.3 |
| GSE30217 | Leukemia | 54 | 964,431 | 1380 | 5 | 190.2 |
| GSE33848 | Lung cancer | 30 | 1,852,426 | 843 | 7 | 215.5 |

Table 2: Results on high-dimensional gene expression data sets.

CASC5, CBX3, CCNB2, CCT5, CDC14B, CDCA7, CEP55, CHRDL1, CIDEA, CKLF, CLEC3B, CLU, CNIH4, DBR1, DDX39, DHRS4, DKFZp667G2110, DKFZp762E1312, DMD, DNMT1, DTL, DTX3L, E2F3, ECHDC2, ECHDC3, EFCBP1, EFHC2, EIF2AK1, EIF2C2, EIF2S2, Ells1, EPHX2, EPRS, ERBB4, FAM107A, FAM49B, FARP1, FBXO3, FBXO32, FEN1, FEZ1, FKBP10, FKBP11, FLJ11286, FLJ14668, FLJ20489, FLJ20701, FLJ21511, FMNL3, FMO4, FNDC3B, FOXP1, FTL, GEMIN6, GLT25D1, GNL2, GOLPH2, GPR172A, GSTM5, GULP1, HDGF, HIF3A, HLA-F, HLF, HNRPK, HNRPU, HPSE2, HSPE1, ILF3, IPO9, IQGAP3, K-ALPHA-1, KCNAB1, KDELC1, KDELR2, KDELR3, KIAA1217, KIAA1715, LDHD, LOC162073, LOC91689, LRRFIP2, LSM4, MAGI1, MORC2, MPPE1, MSRA, MTERFD1, NAP1L1, NCL, NDRG2, NME1, NONO, NOX4, NPM1, NR3C2, NRP2, NUSAP1, P53AIP1, PALM, PAQR8, PDIA6, PGK1, PINK1, PLEKHB2, PLIN, PLOD3, PPAP2B, PPIH, PPP2R1B, PRC1, PSMA4, PSMA7, PSMB2, PSMB4, PSMB8, PTP4A3, RBAK, RECK, RORA, RPN2, SCNM1, SEMA6D, SFXN1, SHANK2, SLAMF8, SLC24A3, SLC38A1, SNCA, SNRPB, SNX10, SORBS2, SPP1, STAT1, SYNGR1, TAP1, TAPBP, TCEAL2, TMEM4, TMEPAI, TNFSF13B, TNPO1, TRPM3, TTK, TTL, TUBAL3, UBA2, USP2, UTP18, WASF3, WHSC1, WISP1, XTP3TPA, ZBTB12, ZWILCH.

After learning the DAG, we computed how many nodes are reachable from each of the 187 nodes. We found out that the gene CCNB2 reaches the greatest number of nodes among all genes (163 nodes). Interestingly, this gene was independently found to be associated with an unfavorable outcome for breast-cancer patients in treatment [35]. As specifically mentioned by [35] "findings suggest that cytoplasmic CCNB2 may function as an oncogene and could serve as a potential biomarker of unfavorable prognosis over short-term follow-up in breast cancer".

### B.6  Learning GBNs using marginal variance

To ensure that the class of GBNs used in our synthetic experiments were non-trivial: meaning the marginal variance of the nodes did not give away the causal ordering, we tested another algorithm, which we will call the marginal-variance algorithm, to compute the DAG order by simply sorting the nodes according to their marginal variance. Figure 4 shows the probability of successful structure recovery across 30 randomly sampled GBNs, for the marginal-variance algorithm. We can observe that the marginal-variance algorithm fails to recover the DAG structure much more frequently as the number of variables grows. At $p = 200$, the algorithm fails to recover the true structure $50\%$ of the time.

## Appendix C  Discussion

### C.1  Using RESIT for learning linear Gaussian SEMs

**Proposition 1.** *Let* $(\mathsf{G}, \mathcal{P}(\mathsf{W}, \mathsf{S}))$ *be a GBN and* $\mathbf{X} \in \mathbb{R}^p$ *be a data sample drawn from* $\mathcal{P}$. *For any variable* $i$, *let* $\boldsymbol{\theta}_i^* = \min_{\boldsymbol{\theta} \in \mathbb{R}^{(p-1)}} \frac{1}{2}\mathbb{E}\left[(X_i - \boldsymbol{\theta}^T \mathbf{X}_{-i})^2\right]$, *and let* $R_i = X_i - (\boldsymbol{\theta}_i^*)^T X_{-i}$ *be the* $i$-th *population residual. Then, the residual* $R_i$ *is independent of* $X_j$ *for all* $j \in -\mathsf{i}$, *i.e.,* $\mathrm{Cov}\left[R_i, X_j\right] = 0$.

Figure 4: Performance of the marginal-variance algorithm that uses sorting of the nodes by marginal variance to learn the DAG order.

A consequence of the above proposition is that, RESIT, which identifies terminal vertices, and subsequently the DAG order, by performing independence tests between the residual $R_i$ and the covariates $X_{-i}$, does not work even in the population setting.

*Proof of Proposition 1.* Without loss of generality, let us write the joint distribution of $(X_i, X_{-i})$ as follows:

$$\begin{pmatrix} X_{-i} \\ X_i \end{pmatrix} \sim \mathcal{N}\left(\mathbf{0}, \begin{pmatrix} \mathbf{A} & \mathbf{b} \\ \mathbf{b}^T & c \end{pmatrix}\right).$$

Then, from standard results for ordinary least squares, we have that $\boldsymbol{\theta}_i^* = \operatorname{argmin}_{\boldsymbol{\theta} \in \mathbb{R}^{p-1}} \mathbb{E}\left[\frac{1}{2}\|\mathbf{X}_{*,i} - \mathbf{X}_{*,-i}\boldsymbol{\theta}\|_2^2\right] = \mathbf{A}^{-1}\mathbf{b}$. Let $R_i = X_i - \mathbf{b}^T\mathbf{A}^{-1}X_{-i}$. Using the fact that both $R_i$ and $X_{-i}$ are mean 0, we get:

$$\begin{aligned} \operatorname{Cov}[R_i, X_{-i}] &= \mathbb{E}\left[R_i X_{-i}^T\right] - \mathbb{E}[R_i]\mathbb{E}\left[X_{-i}^T\right] \\ &= \mathbb{E}\left[X_i X_{-i}^T\right] - \mathbb{E}\left[\mathbf{b}^T\mathbf{A}^{-1}X_{-i}X_{-i}^T\right] \\ &= \mathbf{b}^T - \mathbf{b}^T\mathbf{A}^{-1}\mathbf{A} = \mathbf{0}. \end{aligned}$$

$\square$

## C.2 Computational Complexity

The computational complexity of our algorithm is dominated by the inverse covariance estimation step. As described in [29], the CLIME estimator of the inverse covariance matrix can be obtained by solving $p$ linear programs, each with $2p$ inequality constraints in a $4p$-dimensional vector space. Each of these linear programs can be solved in polynomial time by using interior point methods. Further, state-of-the-art methods for inverse covariance estimation can potentially scale to a million variables [25]. After estimating the inverse covariance matrix, our algorithm performs $(p-1)$ OLS computations in (at-most) $\mathbb{R}^k$, to learn the DAG order and another $(p-1)$ OLS computations to learn the structure and parameters. This can be accomplished in $\mathcal{O}(pk^3)$ time by directly inverting (at-most) $k \times k$ symmetric positive-definite matrices. Thus, it is safe to conclude that our exact algorithm for learning equal noise-variance GBNs is highly scalable.