[Reviews · NeurIPS 2017]

Reviewer 1



[After-rebuttal edit begins] I read the authors' response, and thank them for clarifying many of the issues I raised. However, the authors dismissed my concerns that the quantities in the characterization of the sample complexity are independent of the dimensions (p and k) of the problem. Basically, the authors are saying this. Let's look at an asymptotic regime where we have a sequence of graphs with growing p and k, but where: (1) alpha (restricted faithfulness), (2) the largest effective influence, (3) the least eigenvalue of the Markov-blanket covariance matrix, (4) the largest diagonal entry of the covariance matrix, and the (5) largest entry of the precision matrix, are all constant (or more generally upper/lower bounded depending on whether they're in the numerator/denominator), in this asymptotic regime, the sample complexity is order of k^4 log(p). My concern was, are such limits even possible? If so, are they meaningful? So I took a little bit of a closer look. Based on Lemma 5, to enforce (2)-(5) the weights have to remain roughly constant, and at least bounded from above. Then, based on the proof of Proposition 1, I looked at what kind of alpha we would get when fixing w. And I found that unless the number of children remains bounded (as k grows, mind you!), then alpha will tend to zero with k. The good news is that it still seems polynomial, but this certainly goes against the advertised claim. I don't think this is acceptable. This is not a new issue that I'm raising, I gave the authors a chance to respond to my concern in this rebuttal. In this light, I am forced to lower my recommendation for this paper, despite my general appreciation of results of this nature (constraints that lead to polynomial sample/time learning). [After-rebuttal edit ends] Summary This paper considers the problem of recovering the structure and parameters of Gaussian Bayesian Networks. In particular, it establishes that as long as noises are homoscedastic, then under a milder minimality/faithfulness assumptions it is possible to efficiently recover the GBN. Clarity The paper is heavy on notation, but everything is explained and organized clearly. Some suggestions. Line 43, move the assumption before the sample complexity statement on Line 40, so that \alpha has meaning. Line 130, elaborate on how effective influence governs the Markov blanket property. In Definition 2, Line 137, say that the key is the reverse direction, in turn relaxed in Definition 3. Use Line 157 to introduce the term "homoscedastic" before it's use later. In Assumption 2, would it be better to index and write \kappa_i(\alpha)? In Line 196 the precision matrix is defined without calling it inverse covariance matrix, but then we only call it inverse covariance matrix. Line 199, say We have that (if)... (then). Line 204, say is defined as instead of the def symbol. Significance Recovering parsimonious probabilistic models of data will always be a problem of importance. Finding among such models those for which we can have provable guarantees is necessary when we want to have confidence in the models that we produce. In this vein, this paper succeeds in giving weaker conditions and an algorithm, thus stronger results, to learn a rather rich subset of probabilistic models that are generally notoriously difficult to learn. Flaws I have not found major flaws, but there are many details that need to be clarified in the main body of the paper. Many of these are about the dependence between the meta-parameters of the problem. - Is the only role of k in bounding the size of non-zero elements of columns of the precision matrix? Does it influence anything else? - Proposition 1 doesn't make it clear what the relationship is betwen the structural properties of G, W, and \sigma^2 on one hand, and the resulting \alpha on the other. Would \alpha end up depending on p and k, for example? - Methods that rely on conditional independence tests may require strong faithfulness to analyze, but do we know that they will fail without it? (Line 181) - Is it evident that p does not influence the induced norm of the precision matrix? (Remark 2) - Also, couldn't the infinity norm of the regressors and the least eigenvalue of the Markov-blanket covariance, depend on p and k? (Lemma 3) - I see that Theorem 1 is particularly careful in keeing all the terms I'm asking about explicit. But elsewhere in the paper the operator norm is capped by k only, and no mention is made about the regressor, etc. A clarification is needed. - I would relate the terminal node selection procedure by the max/min thresholding used in the algorithm, back to the property stated in Lemma 1 (Lines 270-271). Because whereas for terminal nodes we know the behavior, it has not been characterized for non-terminal nodes, and the implicit assumption being the ratio would be larger for those. - Is the actual dependence on k conjectured to be k^2 (re: Fig 2)?

Reviewer 2



This paper concerns learning sparse Gaussian Bayesian networks with equal noise variance. The paper shows that, under certain assumptions, a polynomial-time algorithm can recover such network in high probability using a polynomial number of samples. Most learning problems in Bayesian networks are NP-hard and thus it is a pleasure to see variants that are learnable in polynomial time. The paper was quite technical and I was not able to check correctness of the results. Theorem 1. The theorem says that the guarantees hold for some choices of delta and alpha. Does this mean that it is not enough to choose arbitrary delta and alpha and then collect so many samples that the inequality is satisfied? Line 63: Note that the size of the search space alone does not imply that the search is prohibitive. Indeed, the search space of the presented algorithm is also super-exponential.

Reviewer 3



Summary: The paper considers a Gaussian Bayesian Network which is directed acyclic and is governed by structural equations which are linear with additive gaussian noise of identical variance. This class of bayesian networks have been shown to be identifiable in prior work. There has been no polynomial time algorithm known even for this case. The authors propose a polynomial time algorithm which involves inverse covariance estimation, ordinary least squares and a simple identification rule for terminal vertices (leaf vertices in the Directed Acyclic Graph) for this problem. Under the assumption of restricted strong adjacency faithfulness (RSAF) (which essentially says that for all induced sub graphs, variables in the Markov Blanket exhibit high enough influence on a variable) the algorithm identifies the correct graph with O(k^4 log p) samples where k is the maximum size of the Markov Blanket. Strengths: a) As far as I am aware this is the first fully polynomial time algorithm for the problem under this setting. b) The paper hinges on few important observations: 1) Learning undirected gaussian graphical models is very very efficient due to large number of papers on inverse covariance estimation. 2) But usually inverse covariance estimation can only recover the Markov Blanket of every variable (considering the moralized graph) and cannot in general reveal directions. 3) However, if a topological ordering of the vertices respecting the partial order in the DAG is known, then Ordinary least squares can be repeatedly used to figure out parents of a variable i by considering an induced graph with vertices in the ordering upto i and the inverse covariance structure at every subsequent stage. 4) The authors exploit the following key fact construct a topological ordering by discovering terminal vertices (lead vertices) recursively: When a terminal vertex i is regressed on the rest of the nodes, then ratio of regression coefficient of vertex j in the graph to (i,j)th entry in the inverse covariance matrix equal to the noise variance in the problem. In fact, for non terminal vertices, the ratio is strictly larger under RSAF condition. This involves only inverse covariance estimation which is efficient. c)) Because the authors define effective influence characterizing the markov blanket in terms of the weights in the structural equations, authors can handle cases even when the model is not causally faithful (this is a standard assumption for most causal inference algorithms). d) In all its a really good paper. The authors have also compared it to some state of the art algorithms like PC, MMHC and GES and shown that for synthetic gaussian models, they do better in terms of recovery. However, I only have some semi-mild to some mild comments. Comment d is the most serious. It would be great if authors can address it. Weaknesses: a) In page 3 at the end, Markov Blanket of vertex i is defined using effective influence and the information theoretic definition that a set of nodes such that conditioned on them, the rest of the variables are independent to i. Under casual faithfulness, markov blanket is simply - parents, children and co-parents. I think defining it in terms of effective influence lets them deal with pathological cases not covered by causal faithfulness. This lets them work on examples in page 5. I think this must be made clear. Because a lot of readers of BN literature may be used to a graph theoretic notion of Markov Blanket. b) In the example in page 5 below Proposition 1, it appears the for i,j which are neighbors if one chooses a pathological set of weights, only then effective influence becomes zero which can still be handled by the algorithm of the authors. I find it to be a corner case although I agree technically its a super set. The reason for this comment is most other algorithms like PC, MMHC are completely non parametric as far as at least skeleton + v-structure discovery is concerned. But this algorithm has very tight parametric assumptions (so it may not be that fair a comparison). c) Page 2, before Section 2 "MMHC works well in practice but inherently a heuristic algorithm" - I partially agree with this. MMHC has two parts - one is skeleton recovery (which is provably correct under infinite samples) and the other part used score based optimization heuristic to orient. So it may not be totally correct to say it is heuristic. Another minor point - Like 74 , complexity of PC is said to be p^k. k in the paper is Markov Blanket size. However the PC's complexity exponent is the max degree in the graph. Markov blanket is much larger. d) Slightly serious concern: Page 7 , Algorithm 1, Line 24: Now the algorithm after identifying the order is considering induced subgraphs of first i vertices in the ordering and using regression of i on the remaining is finding the parents of i. First, to find the markov blanket of i i the induced sub graph , \hat{\Omega}_{z_i,j} is used - It should be z_i,z_j in the subscript (I think). But the major concern is which inverse covariance is it ? Is it the induced subgraph's inverse covariance upto vertex i in the ordering ? or is it the inverse covariance of the entire graph estimate at 6? Please clarify. This seems important because this step of identifying node i's parents from just from the ordering and regression is something crucial. In fact, the explanation I understood is: the global invariance covariance matrix's markov blanket of i is intersected with the the induced sub graph of vertices till i, now the node i is regressed on its non zero entries. And it will turn out that the parent set of i is correctly identified. This is because in the induced sub graph, markov blanket of i will not include the co-parents. This has been explained in the last but one paragraph in page 14 in Appendix A for proof of theorem 1. But I think this reasoning should be included in the main paper and Line 24 in Algorithm 2 must be clarified even better. e) In Appendix B.5 authors talk about some ref 31 which can split the graph into connected components. Does it mean that the graph in that table has connected components of size at most 500. So has the algorithm applied after the connected components found out ?? If this is the case 500 is not a large number for MMHC algorithm. In fact in the 2006 paper they claim it can actually process 1000 node graphs. Why did the authors not compares to that in that case ? f)Line 100, (i,j) seems to be directed edge from j to i. From my experience, this seems like an odd choice. It is usually i->j. But this is just my personal quip. g) Page 11 . Appendix, Proof of Lemma 1, First line. "forward direction follows from (5)". Should it not be Lemma 5 ??